# Spatially and temporally resolved ice loss in High Mountain Asia and the Gulf of Alaska observed by CryoSat-2 swath altimetry between 2010 and 2019

5 Livia Jakob[1], Noel Gourmelen[1,2,3], Martin Ewart[1], Stephen Plummer[4]

[1]Earthwave Ltd, Edinburgh, EH9 3HJ, United Kingdom
[2]School of GeoSciences, University of Edinburgh, Edinburgh, EH8 9XP, United Kingdom
[3]IPGS UMR 7516, Université de Strasbourg, CNRS, Strasbourg, F-67000, France
[4]European Space Agency, ESA-ESTEC, Noordwijk, 2201 AZ, Netherlands

10 *Correspondence to*: Livia Jakob (livia@earthwave.co.uk)

**Abstract.** Glaciers are currently the largest contributor to sea level rise after ocean thermal expansion, contributing ~30% to sea level budget. Global monitoring of these regions remains a challenging task since global estimates rely on a variety of observations and models to achieve the required spatial and temporal coverage, and significant differences remain between current estimates. Here we report the first application of a novel approach to retrieve spatially-resolved elevation and mass change from radar altimetry over entire mountain glaciers areas. We apply interferometric swath altimetry to CryoSat-2 data acquired between 2010 and 2019 over High Mountain Asia (HMA) and in the Gulf of Alaska (GoA). In addition, we exploit CryoSat's monthly temporal repeat, to reveal seasonal and multiannual variation in rates of glaciers' thinning at unprecedented spatial detail. We find that during this period, HMA and GoA have lost an average of $-28.0 \pm 3.0$ Gt yr$^{-1}$ ($-0.29 \pm 0.03$ m w.e. yr$^{-1}$) and $-76.3 \pm 5.7$ Gt yr$^{-1}$ ($-0.89 \pm 0.07$ m w.e. yr$^{-1}$) respectively, corresponding to a contribution to sea level rise of 0.078 $\pm 0.008$ mm yr$^{-1}$ ($0.051 \pm 0.006$ mm yr$^{-1}$ from exorheic basins) and $0.211 \pm 0.016$ mm yr$^{-1}$. Glacier thinning is ubiquitous except for the Karakoram-Kunlun region experiencing stable or slightly positive mass balance. In the GoA region, the intensity of thinning varies spatially and temporally with acceleration of mass loss from $-0.06 \pm 0.33$ m yr$^{-1}$ to $-1.1 \pm 0.06$ m yr$^{-1}$ from 2013 which correlates with the strength of the Pacific Decadal Oscillation. In HMA ice loss is sustained until 2015-6, with a slight decrease in mass loss from 2016, with some evidence of mass gain locally from 2016-17 onwards.

## 1  Introduction

Glaciers store less than 1% of the mass (Farinotti et al., 2019) and occupy just over 4% of the area (RGI Consortium, 2017) of global land ice, however their rapid rate of mass loss has accounted for almost a third of the global sea level rise during the 21st century (Gardner et al., 2013; WCRP Global Sea Level Budget Group, 2018; Wouters et al., 2019; Zemp et al., 2019), the largest sea level rise (SLR) contribution from land-ice (Bamber et al., 2018; Slater et al., 2021). The quantification of mass

loss in glaciers has posed scientific challenges, resulting in the need to combine various types of observation, and the need to reconcile results obtained using different methods (Gardner et al., 2013). The traditional approach (glaciological method) extrapolates in situ observations (Bolch et al., 2012; Cogley, 2011; Yao et al., 2012; Zemp et al., 2019), however measurements are sparse and possibly biased towards better accessible glaciers located at lower altitudes (Fujita and Nuimura, 2011; Gardner et al., 2013; Wagnon et al., 2013). In contrast, geodetic remote sensing methods rely on comparisons of topographic data or gravity fields to determine glacier changes. Recent geodetic remote sensing methods include (1) Digital Elevation Model (DEM) differencing (Berthier et al., 2010; Brun et al., 2017; Gardelle et al., 2013; Maurer et al., 2019; Shean et al., 2020), (2) satellite laser altimetry (Kääb et al., 2012, 2015; Neckel et al., 2014; Treichler et al., 2019) and (3) Gravity Recovery and Climate experiment (GRACE) satellite gravimetry (Ciracì et al., 2020; Gardner et al., 2013; Jacob et al., 2012; Luthcke et al., 2008; Wouters et al., 2019).

Besides representing an icon for climate change (Bojinski et al., 2014) and impacting global sea level rise, the retreat and thinning of mountain glaciers also affects local communities (Immerzeel et al., 2020). Glacier retreat introduces substantial changes in seasonal and annual water availability, which can have major societal impacts downstream, such as endangering water and food security for populations relying on surface water (Huss and Hock, 2018; Pritchard, 2019; Rasul and Molden, 2019), or introducing geohazards such as extreme flooding (Guido et al., 2016; Quincey et al., 2007; Ragettli et al., 2016). Despite substantial advances with geodetic remote sensing methods, enhancing the spatial resolution and coverage of ice loss estimates, there is currently no demonstrated operational system that can routinely and consistently monitor glaciers worldwide, especially in rugged mountainous terrain and with the necessary temporal resolution.

Prior to CryoSat-2, radar altimetry has traditionally been limited to regions of moderate topography such as ice sheets. The launch of a dedicated radar altimetry ice mission, CryoSat-2, with a sharper footprint, an improvement in the ability to accurately map the ground position of the radar echoes, and the full use of the returned waveform via swath processing (Gourmelen et al., 2018; Gray et al., 2013; Hawley et al., 2009), has seen a near-global expansion of its application to monitoring ice mass changes beyond the two large ice sheets (Foresta et al., 2016, 2018; Gourmelen et al., 2018; Gray et al., 2015; McMillan et al., 2014b). Over regions of more extreme surface topography however, such as those found in mountain glacier areas, the use of radar altimetry has been prohibited by the large pulse-limited footprint, a limited range window (240 m for CryoSat), and closed-loop onboard tracking used to position the altimeter's range window (Dehecq et al., 2013). Despite these limitations, CryoSat's sharper footprint and interferometric capabilities have led to promising studies over mountain glaciers (Dehecq et al., 2013; Foresta et al., 2018; Trantow and Herzfeld, 2016).

The emphasis in this study is to demonstrate the ability of interferometric radar altimetry to monitor regional mass changes in challenging rugged terrain, despite the abovementioned limitations. For this demonstration, we chose High Mountain Asia (HMA) and the Gulf of Alaska (GoA), two regions with complex terrain which have not been previously monitored with radar altimetry. We use CryoSat-2 swath altimetry to derive elevation and mass changes of mountain glaciers from 2010 to 2019, in addition, we exploit the repeat cycle of CryoSat-2 to generate time series (30 days steps) at sub-regional level, giving new insights into seasonal and interannual changes within the two regions. With this study, we ultimately aim to demonstrate the

potential of interferometric radar systems to contribute an independent observation of ice trends on a global scale and at high temporal resolution.

## 1.1 Study regions

The HMA study area includes the Himalaya, the Tibetan mountain ranges, the Pamir and Tien Shan (regions 13, 14 and 15 of the Randolph Glacier Inventory) and is covered by about 100,000 km$^2$ of glacier area for about 95,500 glaciers (RGI Consortium, 2017). Climatic conditions in HMA are characterised by two main atmospheric circulation systems which impact the distribution of glaciers and glaciological changes; the Westerlies and the Indian monsoon (Figure 1). The Westerlies dominate regions in the northwest (Pamir regions, Kunlun Shan, Tien Shan and western Himalayan mountain range) and are responsible for a large fraction of the precipitation deposited particularly during the winter months (Bolch et al., 2012; Li et al., 2015; Yao et al., 2012). The Indian summer monsoon mainly influences glaciers in southern sub-regions (central and eastern Himalayan mountains, Karakoram, Nyainqêntanglha mountains), with decreasing precipitation northward (Bolch et al., 2012; Yao et al., 2012). In contrast to the monsoonal and westerly regimes, the inner Tibetan Plateau is mainly dominated by dry continental climatic conditions. Various studies have found precipitation increases in the Pamir regions and decreases in the central/eastern Himalayan range, affected by changes in the two atmospheric systems, namely the strengthened Westerlies and the weakening Indian Monsoon (Treichler et al., 2019; Yao et al., 2012). As a result of atmospheric forcing, the vast majority of glaciers in the HMA region have been losing mass during the satellite records (Bolch et al., 2019; Farinotti et al., 2015; Maurer et al., 2019) which has led to widespread glacier slowdown (Dehecq et al., 2019).

The GoA region, which we define to encompass the mountain range stretching along the Gulf of Alaska to British Columbia (Region 1 of the Randolph Glacier Inventory 6.0, excluding Northern Alaska) is covered by approximately 86,000 km$^2$ glacier area for a total of about 26,500 glaciers (RGI Consortium, 2017). The glacierised areas stretch from sea level up to over 6000 m. a.s.l., representing a large variety of different glacier types. 67% of the glacier area are in land-terminating glaciers, 13% and 20% are marine-terminating and lake-terminating respectively (Figure 1). Large glacier-to-glacier variations in mass changes have been reported, which are assumed to be driven by climate variability and heterogeneity of glacier elevation ranges (Larsen et al., 2015). The coastal regions along the Alaskan Gulf experience a maritime climate, with the maximum precipitation occurring on the southern slopes of the Coastal Range (Wendler et al., 2017). These mountain ranges act as barriers for the moist air from the Pacific Ocean resulting in rain shadow, i.e. more continental climate, on their leeward side (Le Bris et al., 2011; Wendler et al., 2017). The Pacific Decadal Oscillation (PDO) is another factor which exercises substantial influence on the climate (Wendler et al., 2017) and glacier behaviour (Hodgkins, 2009) within the GoA region. In general, the positive phase of the PDO relates to higher temperatures and more precipitation (Fleming and Whitfield, 2010), whilst a cooling and decrease of precipitation are observed during its negative phase (Papineau, 2001). However, the effects on precipitation especially are spatially heterogenous (Fleming and Whitfield, 2010). Our study period of 2010 to 2019 contains the 2013-4 change from a negative phase of the PDO to a positive phase, contributing to a substantial increase in temperatures

in Alaska from 2014 onwards (Wendler et al., 2017). As a result of atmospheric and oceanic forcings, glaciers in the GoA region have been losing mass during the satellite records (Arendt et al., 2002; Berthier et al., 2010; Wouters et al., 2019; Zemp et al., 2019).

## 2    Data and Methods

In this section, we give a short overview on the data and methods used in this study. More details can be found in the Supporting Information.

### 2.1    Time-dependent elevation from CryoSat-2 observations

We use observations from the SAR Interferometric Radar Altimeter (SIRAL) onboard the European Space Agency (ESA) CryoSat-2 satellite (Wingham et al., 2006). SIRAL is a beam-forming active microwave radar altimeter with a maximum imaging range of ~15 km on the ground. The sensor emits time-limited *Ku-band* pulses aimed at reducing the footprint to ~1.6 km within the beam. Over land-ice, the sensor operates in synthetic aperture interferometric (SARIn) mode, which allows delay-Doppler processing to generate an along-track footprint of ~380 m, while cross-track interferometry is used to extract key information about the position of the footprint centre. In practice however, footprint size will vary depending on properties such as surface slopes, scattering properties, and distance from the Point-of-Closest-Approach (POCA). CryoSat-2 orbits the Earth with a 369-day near-repeat period formed by the successive shift of a 30-day sub-cycle. The satellite has an inclination of 92°, offering improved coverage of the polar regions. We process *level 1b,* baseline C data and the corrected mis-pointing angle for aberration of light (Scagliola et al., 2018) supplied by the ESA ground segment using a swath processing algorithm (Gourmelen et al., 2018). *Level 1b* data is provided as a sequence of radar echoes along the satellite track, which translates into received power, interferometric phase and coherence waveforms for each along-track location. The conventional *level 1b* data processing method consists in extracting single elevation measurements from the power signal in each waveform that corresponds to the POCA between satellite and the ground. In contrast, swath altimetry exploits the full radar waveform to map a dense swath (~5 km wide) of elevation measurements across the satellite ground track beyond POCA (Foresta et al., 2016, 2018; Gourmelen et al., 2018; Gray et al., 2013; Hawley et al., 2009) providing one to two orders of magnitude more elevation measurements compared with POCA and improving the sampling of topographic lows (Foresta at al., 2016). Because swath processing does not rely on retracking, it can retrieve elevation measurements also for atypical waveforms with no clearly defined leading edge such as those found over complex terrain and where retracking often fails to identify a reliable POCA. This makes the CryoSat-2 sensor at present the only radar altimeter able to survey glaciers at high resolution.

### 2.2    Rates of elevation change maps

Rates of elevation change and mass balance are based on ~25 million swath elevation measurements in the GoA region and ~8 million swath elevation measurements in HMA acquired from mid-2010 to mid-2019. The distribution of elevation

measurements with altitude departs somewhat from the glaciers' hypsometry. Hypsometric representativeness of samples within spatial units is a key requirement for robust glacier trend estimates. A bias in the altitudinal distribution of observations can lead to a bias in the total rate of thinning when integrated over a larger domain as rate of thickness change is often strongly correlated with altitude. Therefore we derive a subset of the time-dependent elevation dataset, removing the impact of such point density biases by filtering out swath measurements so as to match the glacier hypsometry binned using 100m elevation intervals (e.g. Treichler et al., 2019), and generate elevation change and mass change estimates from the reduced sample (Figure S5). We remove data sequentially based on measurement uncertainty. This process reduces our sample size by 15% for the GoA and by 30% for HMA. We then follow a similar approach to Foresta et al. (2016) and Gourmelen et al. (2018), however the lower data density and the complexity of the terrain in the GoA region and in HMA require a slight adaptation of the methodology. We bin the elevation measurements into regions of 100 x 100 km, sufficiently large to contain the necessary number of measurements in each bin to ensure sufficient robustness and representativity. Due to the increased bin size (the pixel size used by Foresta et al. [2016] and Gourmelen et al. [2018] is 1000 m) and the variation of elevations within each bin, the topographic signature cannot simply be modelled and therefore needs to be removed using an auxiliary Digital Elevation Model (DEM) (Kääb et al. 2012). We subtract the TanDEM-X 90m DEM (German Aerospace Center [DLR], 2018), which has a near-complete coverage and is contemporaneous of the CryoSat-2 observations, from the swath elevation measurements. The remaining elevation differences (hereinafter referred to as *elevDiff*) are due to time-dependent elevation change that can be related to glacier thickness change as well as errors in the two data sets, temporal heterogeneity (TanDEM-X is a composite of acquisitions from different years) and differences in penetration between the reference DEM (*X-band*) and the swath elevation measurements. The errors related to the reference DEM will result in an increase in spread of the *elevDiff* measurements and is accounted for in the regression model discussed below.

Rates of elevation change are then calculated for each 100 x 100 km bin individually based on *elevDiff* measurements from mid-2010 to mid-2019. In order to achieve the most robust trends we considered several fitting methods, including ordinary least-square, robust regression (e.g. Kääb et al., 2012, 2015), weighted regression (e.g. Berthier et al., 2016; Foresta et al., 2018; Gourmelen et al., 2018), random sample consensus (RANSAC) and Theil-Sen estimator (e.g. Shean et al., 2020). We found that the best results were achieved with a weighted regression model of the *elevDiff* measurements, similar to the methods of Gourmelen et al. (2018). However, whilst their weights are calculated only according to the power attribute, here we assign each observation a weight based on power and coherence, i.e. measurements with high power and low coherence within the sample will have lower weights assigned (see Supplementary Information S1.1). We exclude solutions that display extremely large variability across various regression models, considering them as unstable solutions results (see Supplementary Information S1.2). When fitting the model, we iteratively exclude measurements that are more than 3σ from the mean distance to the fitted line, until no more outliers are present (e.g. Foresta et al., 2016, 2018). We discard bins that did not fulfil a set of quality criteria based on elevation change uncertainties, temporal completeness, interannual changes and stability of regression results (see Supplementary Information S1.2). The remaining bins covered more than 96% of the total glacierised area in the GoA region, and 88% in HMA. To estimate values for the gaps in our dh/dt map we use the altitudinal distribution of elevation

change rates on a sub-regional level (Moholdt et al., 2010a, 2010b; Nilsson et al., 2015), applying the hypsometric averaging methods described in the Supplementary Information (S1.3).

## 2.3 Mass balance and contribution to sea level rise

To obtain volume changes we use the glacierised area of the Randolph Glacier Inventory (RGI 6.0) (RGI Consortium, 2017).

We assume the standard bulk density of $850 \pm 60$ kg/m$^3$ (Huss, 2013) to convert volume changes to equivalent mass changes. This assumption is considered appropriate for a wide range of conditions and longer-term trends, however, this factor can differ significantly for shorter term periods (<3 years) (Huss, 2013). To obtain a region-wide mass balance, mass changes of each individual bin are summed up. We derive mass balance for the unadjusted (biased) and glacier hypsometry adjusted (unbiased) elevation dataset (see Section 2.2), our final mass balance numbers are for the unbiased case. To generate the

contribution to sea level rise (SLR) we assume an area of the ocean of $361.8 \cdot 10^6$ km$^2$ and consider total contributions from all glaciers and then only those glaciers within exorheic basins in High Mountain Asia, based on the HydroSHEDS dataset (Lehner et al., 2006).

## 2.4 Time series of surface elevation changes

CryoSat-2's monthly repeat cycle provides the opportunity to monitor seasonal as well and multiannual trends of surface

elevation. We therefore generate time series with a monthly step (30 days) and a 3-month (90 days) moving window, using the median of all the *elevDiff* observations (residuals from the reference DEM) within a time period with reference to the first month. Time series are generated on bin size level (100 x 100 km), on sub-regional level (using the RGI sub-regions) and for the full study region. The time series of the 100 x 100 km bins are also used as an additional check of the dh/dt quality (see Supplementary Information S1.2), whilst we exploit the sub-regional and regional time series to analyse spatio-temporal

variability in thickness change across both, the GoA and HMA regions. To generate region-wide time series for HMA and the GoA we use an area-weighted mean of the sub-regional time series. Note that as opposed to the linear rates the regional and sub-regional time series displayed in this publication start in January 2011 (with the earliest data from November 2010 using the 90 days window), since we retrieve less swath measurements for the first few months of CryoSat-2's life cycle, impacting the quality of the time series pre-2011. The time series in this paper end in April 2019, with the latest data from June 2019 due

to the 90 days window.

## 2.5 Uncertainty assessment

The error budget on mass change has three uncertainty sources, which are assumed to be independent and uncorrelated: uncertainty on time-dependent elevation change ($\sigma_{\Delta h}$), uncertainty on glacierised area ($\sigma_A$) and uncertainty on mass-volume conversion ($\sigma_p$).

The rate of elevation change uncertainty for each 100 x 100 km bin is based on the standard error of the regression model. We conservatively use a factor of five (Berthier et al., 2014; Brun et al., 2017) for uncertainties on areas without coverage of swath measurements:

$$\sigma_{\Delta h} = \sigma_{\Delta z}(g + 5u) \,, \tag{1}$$

where $g$ is the proportional coverage of glacierised area at 400-metre postings, $u$ is $(1 - g)$ and $\sigma_{\Delta z}$ is the standard error of the regression. To retrieve the uncertainty on extrapolated bins we calculate the differences of all non-extrapolated bins between elevation changes using the plane fit approach and elevation changes using the hypsometric averaging method. The standard deviation of these differences is the uncertainty on elevation change ($\sigma_{\Delta h}$) for all extrapolated bins. We retrieve an uncertainty on elevation change for extrapolated bins of 0.34 m yr[-1] and 0.47 m yr[-1] respectively for High Mountain Asia and the Gulf of Alaska. To account for errors due to temporal changes in glacier extents and polygon digitization (Shean et al., 2020) we use an error of 10% ($\sigma_A = 0.1A$) on the glacierised area $A$ in a bin, even though the reported uncertainty of the RGI is ~8% (Pfeffer et al., 2014). Assuming independence between the two error components ($\sigma_A, \sigma_{\Delta h}$), volume change uncertainty ($\sigma_{\Delta V}$) of a bin is:

$$\sigma_{\Delta V} = \sqrt{(\sigma_{\Delta h}\, A)^2 + (\sigma_A\, \Delta h)^2} \,, \tag{2}$$

where $\Delta h$ is the elevation change rate of the respective bin. To generate the region-wide volume uncertainty ($\sigma_{\Delta V_{tot}}$) we combine all the values (including extrapolated bins) in quadrature. We use a density uncertainty of $\sigma_p = 60$ kg m[-3], and a density mass conversion of $p = 850$ kg m[-3] (Huss, 2013). The total mass balance uncertainty is:

$$\sigma_{\Delta M_{tot}} = \sqrt{(\sigma_{\Delta V_{tot}}\, p)^2 + (\sigma_p\, \Delta V_{tot})^2} \,, \tag{3}$$

where $\Delta V_{tot}$ is the total volume change for the region.

## 3    Results

### 3.1    Spatial coverage and elevation sampling

Using the theoretical pulse-limited footprint size of CryoSat-2, we derive a total spatial coverage of glaciated regions of 55% in the GoA and 32% in HMA respectively. These values are the combined result of the absence of recorded returns due to orbit separation and onboard-tracking limitation (Dehecq et al., 2013), and of data quality. Given that it is estimated that 40% of HMA glaciers are not sampled due to onboard tracking limitations (Dehecq et al., 2013) we estimate that with an appropriate onboard tracking system, the rate coverage for HMA would be as high as 50%. These values are within the high-end of the range of observational methods (Zemp et al., 2019), whilst generally lower than the coverage provided by high resolution sensors (Brun et al., 2017; Shean et al., 2020). As expected from the relatively large footprint of radar altimeters we observe a positive correlation between spatial coverage and glacier size, we do however observe coverage over all glacier sizes (Figure S6).

We observe a bias of the total number of swath measurements towards higher altitudes (e.g. Figure S5), which can be attributed to the onboard tracking tending to favour elevations closest to the satellite. However, a comparison of the glacier hypsometry and the spatial coverage of our data (Figure S4) shows that we still achieve good coverage at low elevations in both regions. In addition, we interpolate missing data based on the relationship between elevation and elevation changes and therefore still capture the changes in the lower reaches of the HMA and GoA glaciers.

**3.2    Elevation changes and mass balance in High Mountain Asia**

The total HMA mass balance between 2010 and 2019 was $-28.0 \pm 3.0$ Gt yr$^{-1}$ ($-0.29 \pm 0.03$ m w.e. yr$^{-1}$), or $-18.3 \pm 2.3$ Gt yr$^{-1}$ ($-0.38 \pm 0.03$ m w.e. yr$^{-1}$) when including only exorheic basins. This mass loss corresponds to $0.078 \pm 0.008$ mm yr$^{-1}$ SLE, or $0.051 \pm 0.006$ mm yr$^{-1}$ when including only exorheic basins. As expected, we retrieve slightly smaller specific mass losses with differences in the order of 0.05 m w.e. yr$^{-1}$ for the GoA region and 0.02 m w.e. yr$^{-1}$ for HMA without removing the

elevation bias towards higher altitudes. We do not observe any significant changes in the overall spatial pattern of elevation change across the two regions between the unbiased and biased results, which indicates that we still achieve a stable result with the reduced sample. Our maps of surface elevation change show a heterogeneous pattern in the Himalayan range, with a cluster of slightly positive/near balance trends in the Kunlun and Karakoram ranges (Figure 2), the so-called "Karakoram anomaly" (Gardelle et al., 2012; Hewitt, 2005). Another striking feature is the gradient from moderate thinning in Spiti-Lahaul

and western Himalaya ($-0.25 \pm 0.09$ m w.e. yr$^{-1}$) to increasingly negative surface elevation changes along the central ($-0.43 \pm 0.14$ m w.e. yr$^{-1}$) and eastern ($-0.56 \pm 0.16$ m w.e. yr$^{-1}$) Himalayan mountain range, with the Nyainqêntanglha mountains and Hengduan Shan ($-0.98 \pm 0.22$ m w.e. yr$^{-1}$) showing the highest negative trends.

We display the altitudinal distribution of elevation changes in Figure 6 and a comparison with Brun et al. (2017) in Figure S7. While some variability exists along the profiles, in particular over regions and elevation range containing fewer glaciers that

can reflect a less robust solution and or spatial variability in glacier response, trends between elevation and ice thickness change are clearly visible. In general, we observe decreasing negative trends with increasing altitudes, which is an expected pattern (Brun et al., 2017; Gardelle et al., 2013). We find the steepest gradient (Figure 6, S7, Table S4) in the Nyainqêntanglha/Hengduan Shan, which is in line with the findings of Brun et al. (2017). We also observe lower or even inverse gradients in Bhutan/East Himalaya, Spiti-Lahaul/West Himalaya, Karakoram/West-Kunlun and Pamir (Figure 6, S7,

Table S4), which have been reported previously and been related to debris thickness (Bisset et al., 2020; Brun et al., 2017).

We show temporal variability of surface elevation change for the whole HMA region (Figure 4), the RGI second order regions (Figure 5, S1) and the regions by Brun et al. (2017) (Figure S2). The monthly time series show sustained multiannual trends across almost all of the subregions until 2015-6, and decreased loss or even mass gain from 2016/2017 onwards (Figure 5, S2), which is also reflected in the full HMA time series (Figure 4), and consistent with previous observation (Ciracì et al., 2020).

The Karakoram region in particular displays thinning from 2011 to 2014/5 before abating and thickening again from 2016/7. This shift of thinning rates post-2015 is also clearly seen in Bhutan/East Himalaya, Kunlun (West and East), Tien Shan, Pamir Alay/Hissar Alay and Nyainqêntanglha/Hengduan Shan (Figure 5, S1, S2).

### 3.3 Glacier elevation changes and mass balance in the Gulf of Alaska

In general, we find much higher mass losses in the Gulf of Alaska than in High Mountain Asia. Over an area of ~86,000 km$^2$, including all 26,490 glaciers in the RGI region 1 except Northern Alaska, we estimate a total mass balance of –76.3 ± 5.7 Gt yr$^{-1}$ (–0.89 ± 0.07 m w.e. yr$^{-1}$), contributing –0.211 ± 0.016 mm yr$^{-1}$ to global sea level rise. Surface elevation change maps (Figure 3) display an expected pattern with more negative trends towards lower elevations close to the coast. Note that some of the lower rates observed in the St Elias Mountain are likely the result of the presence of accumulation areas of large glaciers e.g. Hubbard and Bering glaciers in these particular grid cells. We present sub-regional estimates aggregated on the RGI 6.0 second order regions in Table 2. The largest mass loss is seen in the Northern Coast Ranges (–1.08 ± 0.09 m w.e. yr$^{-1}$; –24.8 ± 2.1 Gt yr$^{-1}$) and Saint Elias Mountains (–1.03 ± 0.10 m w.e. yr$^{-1}$; –34.1 ± 3.4 Gt yr$^{-1}$), especially the Yukutat and Glacier Bay region, which is in line with the spatial patterns of Luthcke et al. (2008) and Luthcke et al. (2013). The lowest thinning rates are observed in the Alaska Range mountains (–0.41 ± 0.05 m w.e. yr$^{-1}$), which is also in agreement with other studies (Berthier et al., 2010; Luthcke et al., 2008). We observe a clear correlation between surface elevation changes and altitude (Figure 11, Table S2), with the highest negative trends at low altitudes in the Saint Elias Mountains and Coast Ranges.

We display temporal variability of surface elevation change for the whole GoA region (Figure 4), the RGI sub-regions (Figure 9, S3) and for different elevation bands within sub-regions (Figure 10). Figure 9 shows negative trends across all the sub-regions. The four coastal sub-regions – Alaska Pena, Western Chugach Mountains, Saint Elias Mountains and Coast Ranges – display a seasonal oscillation, with an annual surface elevation maximum in spring and annual surface elevation minimum in autumn. In contrast, the seasonal cycle of Alaska Range mountains is shifted, with the thickness maximum in winter, which is also somewhat visible in the time series by Luthcke et al. (2008). A very noticeable feature within the full GoA time series is the acceleration of thinning from 2013-4 onwards (Figure 4). We record an acceleration of thinning from –0.06 ± 0.33 m yr$^{-1}$ (January 2011 to January 2013) to –1.1 ± 0.06 m yr$^{-1}$ (January 2013 to January 2019). We observe this almost consistently across the five sub-regions, but this is most pronounced in the Saint Elias Mountains, the Western Chugach mountains and Coast Ranges.

## 4 Discussion

### 4.1 Uncertainty

While our uncertainty methods follow existing approaches and our error bounds are similar in magnitude to Brun et al. (2017), Kääb et al. (2012) and Shean et al. (2020) but lower than GRACE-based estimates, several additional potential sources of errors could impact the results. Radar altimetry, and delay-doppler radar in particular, has been shown to be sensitive to surface slopes, and in particular to slope in the direction of the satellite's flight path, in regions like HMA and GoA this impact will also be seen in the performance of the onboard tracker as for large slopes the system is expected to "lose lock". While we do observe a decreased coverage compared to other, less mountainous, glaciated regions, we also demonstrated here that

measurements do cover the entire elevation range of glaciers in the HMA and GoA regions allowing us to match the glaciers' hypsometry. We also do not observe significant coverage bias in function of glacier orientation with respect to the satellite's track path. The spatial coverage is such that we demonstrably resolve spatial, altitudinal, and temporal evolution of glacier elevation.

It is a well-known observation that microwave pulses scatter from the surface as well as the subsurface, which can lead to elevation change bias in regions of historically anomalous melt events (Nilsson et al., 2015); or at seasonal time-scale (Gray et al., 2019). Over most regions however, it has been shown that surface elevation change from CryoSat over annual and pluri-annual time scale are consistent with in-situ, airborne, and meteorological observations (Gourmelen et al., 2018; Gray et al., 2015, 2019; McMillan et al., 2014a; Zheng et al., 2018). Using static glacier masks can also lead to errors in regions of rapid dynamic changes. In general, these limitations are known and efforts are currently underway in the community to improve uncertainty analysis, and develop new glaciers outlines products (RAGMAC, 2019).

Although time series are generally reflecting the actual change in surface elevation, there are a number of limitations that are important to keep in mind when interpreting the results from radar altimetry. For the reasons stated above, scattering properties can induce elevation biases at seasonal time-scale (Gray et al., 2019). In addition, integrating changes over large regions can lead to spatial heterogeneity in the successive time steps, in particular when the data volume becomes too low. These limitations may explain some of the observed patterns, and in particular the few cases where seasonal variability is larger than what is expected from our knowledge of SMB in the regions.

## 4.2 High Mountain Asia

### 4.2.1 Temporal variability

The seasonal and annual time series variability reflects the influence of atmospheric circulations and precipitation seasonality in High Mountain Asia on ice thickness change. Sub-regions dominated by winter accumulation (generally westerly regimes), such as the Hindu Kush, Western Himalaya and the Pamir region (Pohl et al., 2015; Yao et al., 2012), show the typical seasonal pattern with mass accumulation during winter/early spring and mass losses in the summer/autumn months (Figure 5). Contrarily, sub-regions such as Central Himalaya, Eastern Himalaya and Hengduan Shan show a more heterogeneous seasonal pattern. The elevation change time series of these three sub-regions indicate that the annual cycle has two maxima, with a first maximum in winter and a second and smaller peak in summer (Figure 5, S1). Receiving summer-accumulation through the Indian monsoon these sub-regions generally have a precipitation maximum in July/August, however they are also defined by a high variability of precipitation regimes (Maussion et al., 2014) and a high temperature range (Sakai and Fujita, 2017), resulting in glaciers with varying types over very short distances (Maussion et al., 2014). The impact of this variability becomes evident when compared to the more periodic seasonal patterns of the Hindu Kush, Western Himalayas and Pamir time series. This also stands in contrast with the inner Tibetan Plateau, dominated by a more continental climate, where glaciers exhibit almost no intra-annual cycle.

In general, the heterogeneity of the time series reflects the sensitivity of mountain glaciers to meteorological patterns and changes and emphasises that glaciers in High Mountain Asia cannot be considered as one entity with uniform temporal variability and sensitivity to changes.

### 4.2.2     Comparison of regional mass balance with previous work

A comparison of mass balance results in the literature indicates that, while all the studies agree on the general trend in mass loss and spatial variability of mass loss, there is large degree of variability between estimates. While some of the variability can be attributed to the diversity of time-span and region boundaries used, there are also clear differences between observation methods (Figure 8a). Note that here we are only comparing region-wide mass trends with the results closest in space and time to this study, whilst sub-regional differences are discussed in the next section.

Our total mass balance of $-28.0 \pm 3.0$ Gt yr$^{-1}$ ($-0.29 \pm 0.03$ m w.e. yr$^{-1}$) is in good agreement with the $-28.8 \pm 12$ Gt yr$^{-1}$ by Ciracì et al. (2020), a study based on the GRACE and GRACE Follow-On mission covering the period of 2002 to 2019. The results are similar to the estimates provided by various ICESat studies for the years 2003 to 2008, including the $-28.8 \pm 2.2$ Gt yr$^{-1}$ by Treichler et al. (2019), the $-24 \pm 2$ Gt yr$^{-1}$ by Kääb et al. (2015) (excludes the Tien Shan and the Inner Tibetan Plateau) and the $-26 \pm 12$ Gt yr$^{-1}$ by Gardner et al. (2013) [based on ICESat and GRACE]. Our estimates are higher than recent DEM differencing studies such as the $-19.0 \pm 2.5$ Gt yr$^{-1}$ ($-0.19 \pm 0.03$ m w.e. yr$^{-1}$) by Shean et al. (2020) and the $-16.3 \pm 3.5$ Gt yr$^{-1}$ ($-0.16 \pm 0.08$ m w.e. yr$^{-1}$) by Brun et al. (2017).

Besides the differences in data and methodology, a part of these disagreements can be explained by the time periods. Maurer et al. (2019) and King et al. (2019) find that the thinning rates in the Himalayas have increased from the interval 1975–2000 to 2000–2016. This trend seems to have continued in more recent years, with Ciracì et al. (2020) observing significant variation in rates of mass loss during the period between 2002 to 2019, with mean rates of loss 35% larger during the CryoSat period than between 2002 and 2010, which could explain our more negative mass balance in comparison to Brun et al. (2017) [2000 to 2016] and Shean et al. (2020) [2000 to 2018].

### 4.2.3     Comparison of sub-regional mass balances with previous work

Our higher regional mass loss when comparing to the two DEM differencing studies by Brun et al. (2017) and Shean et al. (2020) are mostly down to differences in the South-eastern Himalaya – especially Nyainqêntanglha/Hengduan Shan – and in the Pamir regions. We used the regions by Brun et al. (2017), the RGI 6.0 second order regions and the HiMAP regions (Bolch et al., 2019) to compare our results with other estimates (Figure 7, Figure S8, Figure S9, Table S1, Table S2). For a full discussion of regional differences between estimates of recent studies refer to Bolch et al. (2019). Our results are in line with general findings by Bolch et al., (2019) in the sense that we obtain similar results in sub-regions where there is a good agreement in general between studies, such as Tien Shan, Karakoram, West Nepal (West Himalaya) and Hindu Kush. For Nyainqêntanglha (named Hengduan Shan and S & E Tibet in the RGI sub-region masks) – one of the most controversial

regions – Shean et al. (2020) and Brun et al. (2017) report significantly less negative mass trends (–0.50 ± 0.15 and –0.62 ± 0.23 m w.e. yr$^{-1}$) than our estimates of –0.97 ± 0.19 m w.e. yr$^{-1}$, whilst *in situ* measurements (–0.94 m w.e. yr$^{-1}$ by Yao et al.

[2012] based on the Parlung glaciers between 2006 and 2010) and ICESat studies (Kääb et al., 2015; Treichler et al., 2019) find higher negative rates for the survey time period of 2003 to 2008. We also record higher mass losses in Eastern Himalaya / Bhutan, adjacent to the Nyainqêntanglha mountains. The differences in Nyainqêntanglha and Eastern Himalaya between our estimates and the ones of Brun et al. (2017) and Shean et al. (2020) [time periods 2000-2016 and 2000-2018] fit in with the generally observed acceleration of mass loss in South-East Asia over the past decades (Maurer et al., 2019; Zemp et al., 2019).

Some studies suggest the weakening of the Indian summer monsoon as the primary source of increased thinning (Salerno et al., 2015), whilst other studies find no widespread precipitation decrease in monsoonal regimes which could account for all of these changes and attribute the temperature sensitivity of glaciers in monsoon-dominated regions as the main driver (Maurer et al., 2019). In fact, glaciers in Hengduan Shan, Nyainqêntanglha and Eastern Himalaya have been found to exhibit the highest sensitivity towards temperature in the whole HMA region (Sakai and Fujita, 2017).

Contrasting estimates have also been published for the Pamir and Pamir Alay mountains (Hissar Alay), where high (Kääb et al., 2015), moderate (this study; Ciracì et al., 2020; Gardner et al., 2013), slight mass losses (Brun et al., 2017; Shean et al., 2020), and even mass gains (Gardelle et al., 2013) have been reported. Part of the discrepancy can be attributed to time variability in mass loss (Brun et al., 2017) and driven by fluctuation in winter precipitation (Smith and Bookhagen, 2018). CryoSat time series indeed suggest near-balance between 2011 and 2015 and increased mass loss from 2015 onwards, which

could account for the higher mass loss estimates in comparison to the DEM differencing studies covering the last two decades (Brun et al., 2017; Gardelle et al., 2013; Shean et al., 2020).

The spatial thinning pattern in the Kunlun-Karakoram area (Figure 2) confirms the suggestion of previous studies (Brun et al., 2017; Gardner et al., 2013; Kääb et al., 2015), that the so-called "Karakoram anomaly" (Gardelle et al., 2012; Hewitt, 2005) stretches up to West Kunlun Shan, which is considered now the centre of the anomaly. We record less mass gain in Kunlun

(+0.01 ± 0.05 m w.e. yr$^{-1}$; +0.06 ± 0.05 m w.e. yr$^{-1}$ in the Western part of Kunlun) than previous studies – indicating that the Karakoram anomaly might not persist long-term (Farinotti et al., 2020; Rounce et al., 2020). This observation is also reflected in the elevation change profile of the Kunlun regions, where Brun et al. (2017) find constant thickening at almost all elevation during the survey time period of 2000 to 2016, whilst we record thinning at lower elevations (see Figure S7). These findings suggest a shift towards negative mass balance at lower elevations in the Kunlun region in comparison to the previous decade.

However, our time series suggests increased mass gain from 2016 in Western Kunlun, and also mass gain in the Karakoram (Figure 5). At the same time, we also observe decreased thinning rates in Inner Tibet and East Kunlun. These changes could be a short-term trend, however, it displays the limitation of all mentioned studies (including this study) when deriving linear trends in a region like High Mountain Asia with large inter-annual climate variability and associated glacier changes.

We generally find better agreements with Shean et al. (2020) – the study including an additional 2+ years (2017 and 2018) in

comparison to Brun et al. (2017), and thus more closely aligned with our time period. This potentially indicates that a large part of the disagreements could be related to inter-annual variability and survey time period.

### 4.3    Gulf of Alaska

#### 4.3.1    Temporal variability

The increased thinning from 2013-4 onwards (see Figure 4, 9, S3), which we observed across all sub-regions, has also been reported by Wouters et al. (2019) and Ciracì et al. (2020) in GRACE time series covering the whole Alaska region. This change correlates with the change from a negative PDO phase to a positive phase in 2013-4, which resulted in increased temperatures (Wendler et al., 2017). This is in agreement with Wouters et al. (2019), finding that their interannual mass change variations negatively correlate with the May-September PDO and May-August NAO indices. Our findings further suggest that the

sensitivity of glaciers to the 2013-4 temperature change increases towards lower elevations (Figure 10). The fact that the strongest impact is observed in the coastal regions is likely due the higher sensitivity of maritime glaciers in Alaska to temperature change (Gregory and Oerlemans, 1998) and the lower elevations within these regions.

#### 4.3.2    Comparison of total mass balance with previous work

Our total mass budget of $-76.3 \pm 5.7$ Gt yr$^{-1}$ ($-0.89 \pm 0.07$ m w.e. yr$^{-1}$) agrees with existing estimates, including those using

GRACE ($-76 \pm 4$ Gt yr$^{-1}$, $-72.5 \pm 8$ Gt yr$^{-1}$ and $-69 \pm 11$ Gt yr$^{-1}$ by Sasgen et al., 2012, Ciracì et al., 2020 and Luthcke et al., 2013) and ICESat ($-65 \pm 12$ Gt yr$^{-1}$ by Arendt et al., 2013) as well as a study from airborne altimetry ($-75 \pm 11$ Gt yr$^{-1}$ by Larsen et al., 2015) and a consensus estimate combining glaciological and geodetic observations ($-73 \pm 17$ Gt yr$^{-1}$ / $-0.85 \pm 0.19$ m w.e. yr$^{-1}$ by Zemp et al., 2019) (Figure 8b). Our result is significantly more negative than two GRACE studies, with estimates of $-53 \pm 14$ Gt yr$^{-1}$ (Wouters et al., 2019) and $-42 \pm 6$ Gt yr$^{-1}$ (Jacob et al., 2012). Besides the variations in

methodologies and data between these studies, also differences in study area extents, glacier masks and volume to mass conversion factors contribute to the spread of total mass change results. Our estimates correspond to the RGI 1 region (excluding Northern Alaska) to make the results more comparable for future studies. In general, our total mass balance is more negative than most other studies' findings, reflecting the increased thinning rates we show in the sub-regional time series from 2013-4.

#### 4.3.3    Comparison of sub-regional mass balances with previous work

Since there is no prevalent sub-region mask used by more recent studies, we cannot directly compare and validate our results on a sub-regional level. Mass balance or surface elevation change estimates that overlap with our time period are either spatially not resolved (e.g. Gardner et al., 2013; Zemp et al., 2019), presented on a glacier to glacier basis (e.g. Larsen et al., 2015) or GRACE mascon extents (e.g. Luthcke et al., 2008, 2013).

Figure 12 displays a comparison with the 1962 to 2006 estimates of Berthier et al. (2010), providing insights into changes of thinning rates since this time period. Our results are consistently more negative, however the general pattern with the lowest

changes discovered in Alaska Range and the highest rates taking place in the Coast Ranges is in agreement with Berthier et al. (2010). We see the largest differences along the east coast − particularly in the Saint Elias mountains − which are also the areas where the lowering of mean surface elevations after 2013-4 has been most pronounced (see Figure 9, S3). In comparison to the study by Berthier et al. (2010), which is based on sequential Digital Elevation Models over the time period from 1962 to 2006, we observe similar elevation changes at the lowest altitudes but less steep gradients in sub-regions along the east coast (Figure 11, Table S2). This is particularly pronounced in the Saint Elias Mountains, where Berthier et al. (2010) show near-balance at around 1000 metre above sea level, whilst our estimates suggest a surface elevation change of $-1.5$ m yr$^{-1}$ at the same altitude. These findings indicate a propagation of thinning upstream since 1962 to 2006. In contrary, whilst on the Alaska Peninsula elevation changes have increased at lower altitudes, the limit of the thinning area has stayed the same since the survey time period of Berthier et al. (2010).

In the Western Chugach Mountains, Alaska Range and Alaska Peninsula we observe a decrease of thinning rates towards the lowest elevations of these sub-regions, which can be attributed to the effect of debris cover and the temporal evolution of glacier extent during the study period, one of the limitations of using static glacier masks. This characteristic has been observed, although more pronounced and across all sub-regions, by Berthier et al. (2010) and Arendt et al. (2002).

## 5    Conclusion

We exploit CryoSat-2 interferometric swath processed data from 2010 to 2019, with a total of 33 million elevation observations, to generate new and independent mass balance estimates for two mountain regions, the Gulf of Alaska (GoA) and High Mountain Asia (HMA). We also generate observations at sub-regional level and extract elevation-dependant thinning rates, revealing contrasting mass loss across sub-regions. Finally, we extract monthly time series of elevation change, exploiting CryoSat's high temporal repeat, to reveal seasonal and multiannual variation in rates of glaciers' thinning. We find that between 2010 and 2019, HMA has lost mass at a rate of $28.0 \pm 3.0$ Gt yr$^{-1}$ ($0.29 \pm 0.03$ m w.e. yr$^{-1}$), and the GoA region has lost mass at a rate of $76.3 \pm 5.7$ Gt yr$^{-1}$ ($0.89 \pm 0.07$ m w.e. yr$^{-1}$), for a sea-level contribution of $0.078 \pm 0.008$ mm yr$^{-1}$ ($0.051 \pm 0.006$ mm yr$^{-1}$ from exorheic basins) and $0.211 \pm 0.016$ mm yr$^{-1}$ respectively for HMA and the GoA. These estimates are broadly consistent with the range of estimates generated by previous studies and highlight the significant discrepancies that remain in the assessments of mass loss for these two regions.

In HMA we find the most negative surface elevation trends in the Nyainqêntanglha mountains, Hengduan Shan, the East Himalayan range and the Tien Shan and slightly positive/near balance trends in the Kunlun and Karakoram ranges, known as the "Karakoram anomaly". The monthly time series of this paper reflect the sensitivity of glaciers in HMA to meteorological patterns and changes and emphasises that the temporal variability of glaciers in High Mountain Asia varies spatially. We show sustained multiannual trends across almost all of the subregions until 2015-6, and decreased loss or even mass gain from 2016/2017 onwards.

Negative mass trends are also observed in all of the sub-regions in the GoA region, with the largest mass losses in the Coast Ranges and the Saint Elias Mountains. The GoA time series reveal an increased mass loss from 2013-4, most pronounced in

sub-regions along the south-central and south-east coast (Saint Elias Mountains, Chugach mountains and Coast Ranges) at lower elevations. This mass loss acceleration is linked with the change from a negative to a positive Pacific Decal Oscillation (PDO) which resulted in increased temperatures. In general, our time series not only display the sensitivity of glaciers to climatic conditions and changes but also illustrate the limitations of linear models when deriving thickness changes, highlighting the importance of higher temporal resolution to generate robust long-term trends.

This is the first study to demonstrate the ability of interferometric radar altimetry to monitor large-scale change in thickness, mass and sea-level contribution of glaciers across regions of extreme topography. This, along with recent work in the Arctic and Patagonia demonstrates the potential of such a system to monitor trends in ice mass on a global scale and with increased temporal resolution. It also demonstrates the ability to monitor monthly change and paves the way to an observation-based quantification of seasonal accumulation and melting processes, a task that will likely require combination with regional climate

models, and with other sensors such as IceSat-2 and high-resolution DEMs.

*Supplement.* The supplement related to this article is available online at: [insert_link]

*Author contributions.* NG and LJ designed the study. LJ performed data analyses and prepared the manuscript. NG generated the swath altimetry dataset and contributed to data interpretation. All authors commented on the manuscript.

*Competing interests.* The authors declare that they have no conflict of interest.

*Acknowledgements.* This work was performed under the European Space Agency's Support to Science Element CryoSat+ Mountain Glacier and we particularly wish to thank Mark Drinkwater and Diego Fernandez at ESA for actively supporting the development of CryoSat beyond the initial primary objectives of the mission. Initial investigations of CryoSat data over High Mountain Asia were performed by Amaury Dehecq. We are grateful to Jonathan Alford for facilitating the data processing and pipeline and to Alex Horton for processing reference DEMs. We further thank ESA for providing free access 470   to CryoSat-2 data, the Randolph Glacier Inventory (RGI) consortium for providing free access to glacier and debris masks, the German Aerospace Center (DLR) for providing free access to the TanDEM-X DEM and NASA for providing free access to the SRTM DEM. We thank Tobias Bolch and an anonymous reviewer, as well as Etienne Berthier, whose constructive comments and suggestions greatly helped to improve this paper.

*Financial support.* This research has been supported by the ESA project CryoSat+ Mountain Glaciers (contract no. 4000114224/15/I-SBo).

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

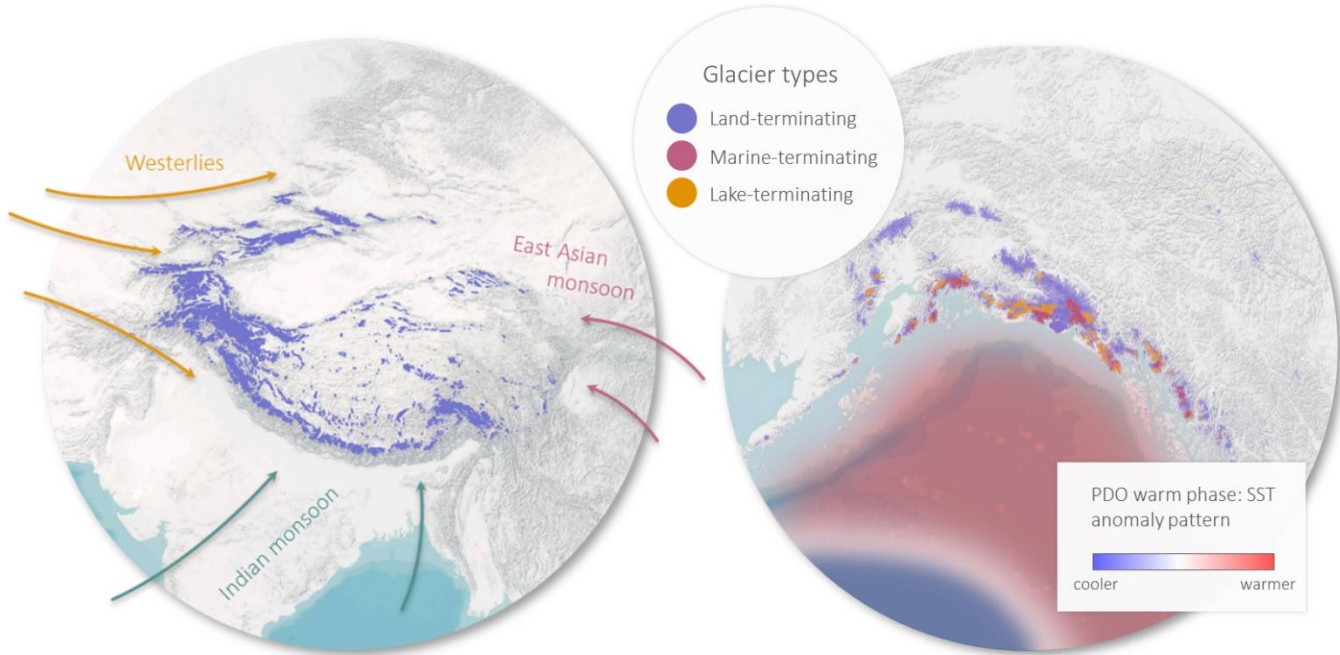


**Figure 1: The two study areas.** *Left*: **High Mountain Asia (HMA) glaciers with arrows showing the main atmospheric circulation systems.** *Right:* **The Gulf of Alaska (GoA) glaciers coloured by glacier type (land-terminating, marine-terminating and lake-terminating). The Sea Surface Temperature (SST) anomaly of the Pacific Decal Oscillation warm/positive phase is displayed in blue-white-red shading.**


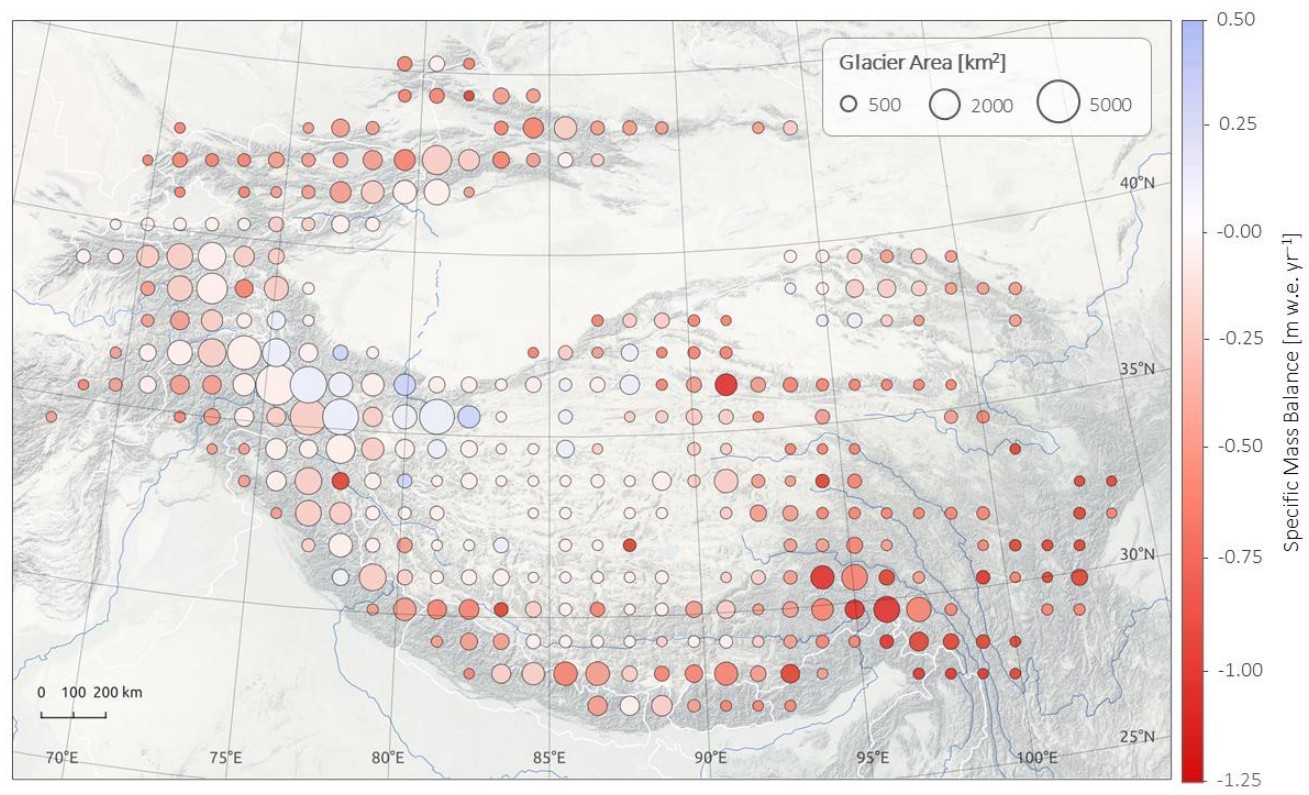

**Figure 2: Specific glacier mass balance (m w.e. yr⁻¹) in High Mountain Asia (HMA) for the period of 2010 to 2019 on a 100 x 100 km grid. The size of the circles is scaled by the total glacierised area within a 100 x 100 km bin.**


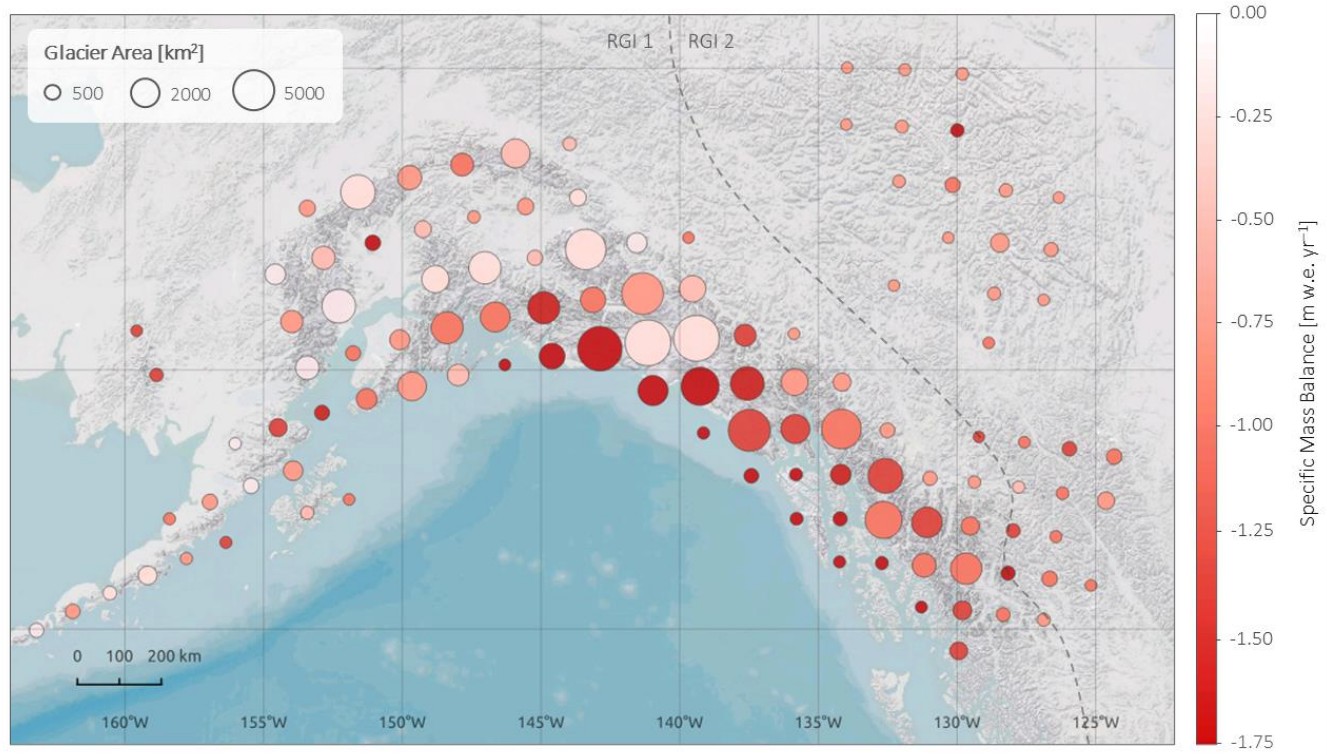

**Figure 3: Specific glacier mass balance (m w.e. yr⁻¹) in the Gulf of Alaska (GoA) for the period of 2010 to 2019 on a 100 x 100 km grid. The size of the circles is scaled by the total glacierised area within a cell. Note that our total mass change estimate of –76.3 ± 5.7 Gt yr⁻¹ (–0.89 ± 0.07 m w.e. yr⁻¹) only include glaciers from the RGI region 1 (Alaska). Including also the Northern Rocky Mountains and the Mackenzie and Selwyn mountains we retrieve a mass change of –77.7 ± 5.7 Gt yr⁻¹.**

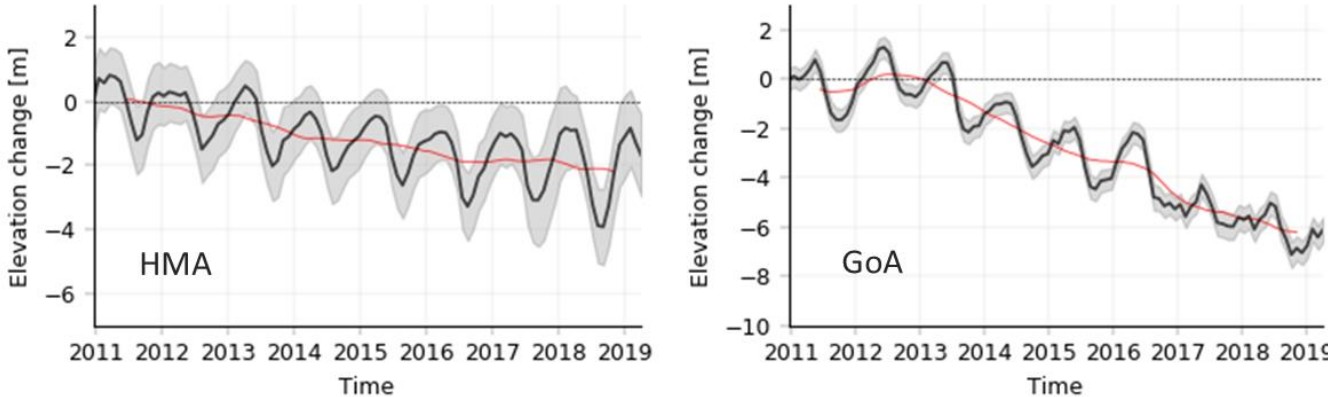

**Figure 4: Monthly surface elevation change time series for High Mountain Asia (*left*) and the Gulf of Alaska (GoA) region (*right*). The grey lines display the elevation change time series with the uncertainty envelope. The red line displays a 12-month moving average.**

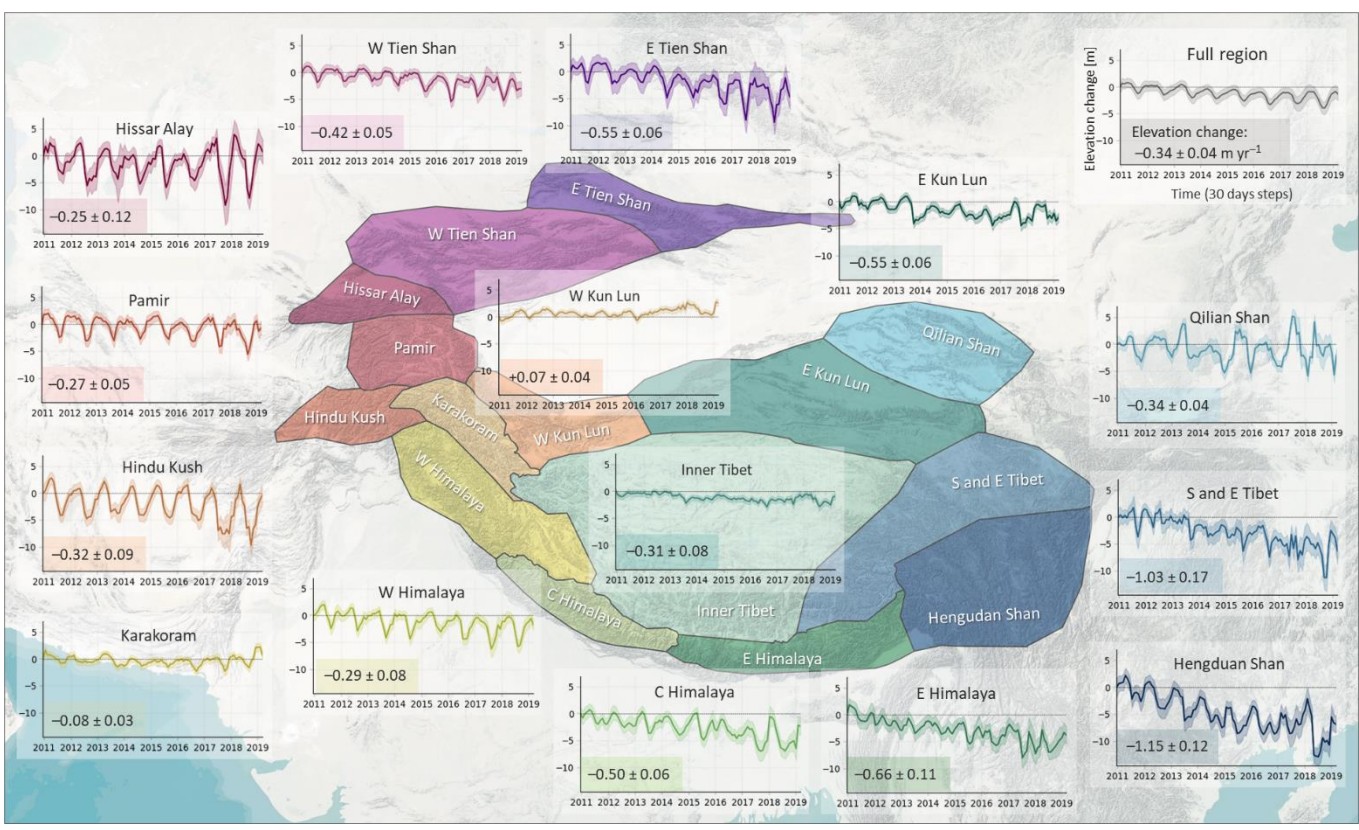

**Figure 5: High Mountain Asia (HMA) 30-day elevation change time series in the RGI 6.0 second order regions. The coloured line displays the time series with the uncertainty envelope (*y-axis*: elevation change [m], *x-axis*: time [30-days steps]). The numbers describe the elevation change with uncertainties in m yr⁻¹.**

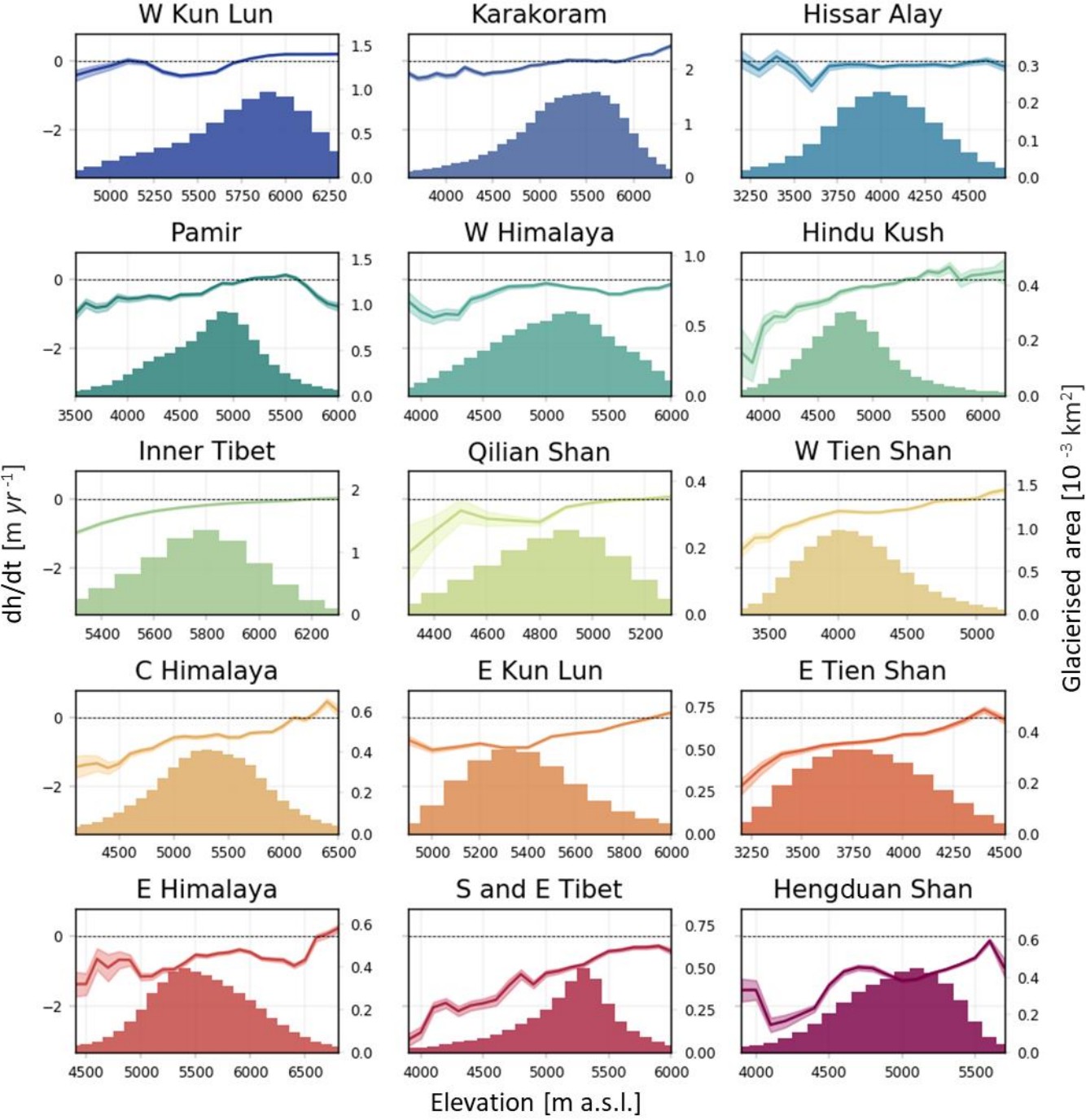

**Figure 6: Altitudinal distribution of elevation changes and glacier hypsometry functions in High Mountain Asia (HMA) in RGI 6.0 sub-regions between 2010 and 2019. The lines show elevation change rates with uncertainty envelopes plotted against 100 m elevation bands (*left y-axis*). The bars display the glacier hypsometry (*right y-axis*).**

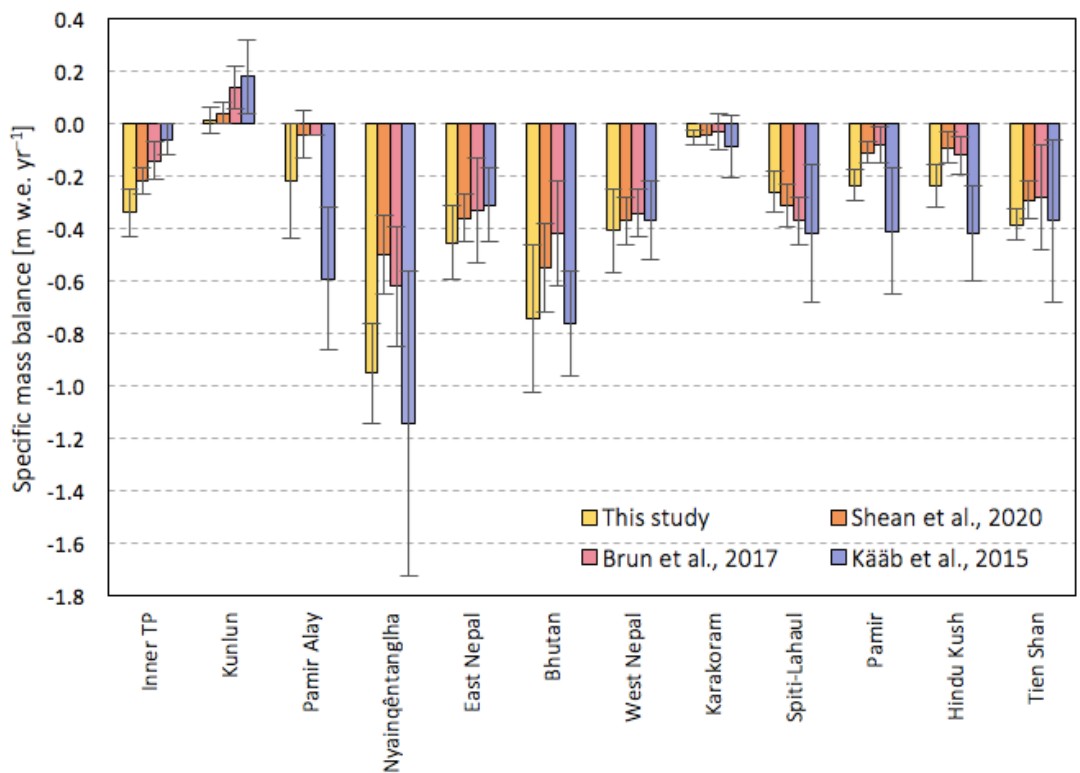


**Figure 7: High Mountain Asia (HMA) specific mass balance trends on a sub-regional level (using the sub-regions of Brun et al. [2017]) in comparison with DEM differencing and ICESat studies. It is important to note that Shean et al. (2020) cover the time period of 2000 to 2018, Brun et al. (2017) cover the time period of 2000 to 2016 and Kääb et al. (2015) cover the time period 2003 to 2008, whilst this study covers the time period of 2010 to 2019. We have complemented the data from Kääb et al. (2015) with ICESat**
**data from Brun et al. (2017) for the sub-regions Kunlun, Inner TP, Tien Shan and Pamir Alay, which extended the estimates of Kääb et al. (2015) using the same method.**

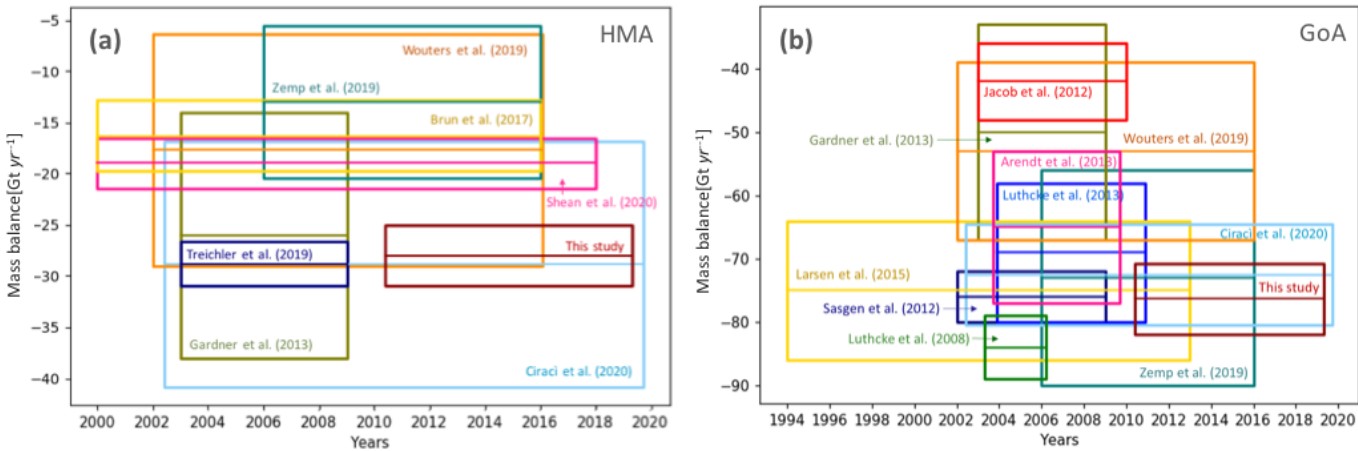

Figure 8: Estimates of mass balance [Gt yr⁻¹] as published in different studies for High Mountain Asia (*left*) and Alaska (*right*).

Figure 9: Gulf of Alaska (GoA) monthly elevation change time series on sub-regional level. The coloured lines display the time series with the uncertainty envelope (*y-axis*: elevation change [m], *x-axis*: time [30-days steps]). The numbers describe the elevation change with uncertainties in m yr⁻¹.

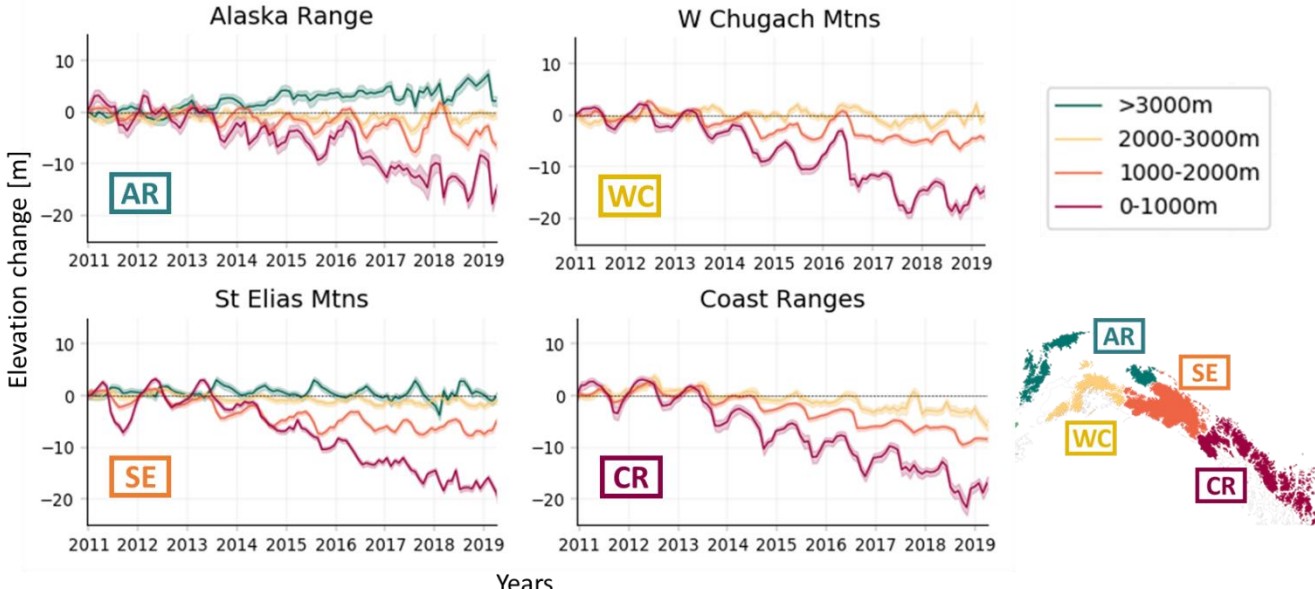

**Figure 10: Gulf of Alaska (GoA) monthly surface elevation change time series with uncertainty envelopes at different elevation bands aggregated on the RGI 6.0 second order regions. The different colours represent the elevation bands (>3000 m, 2000-3000 m, 1000-2000 m, 0-2000 m).**

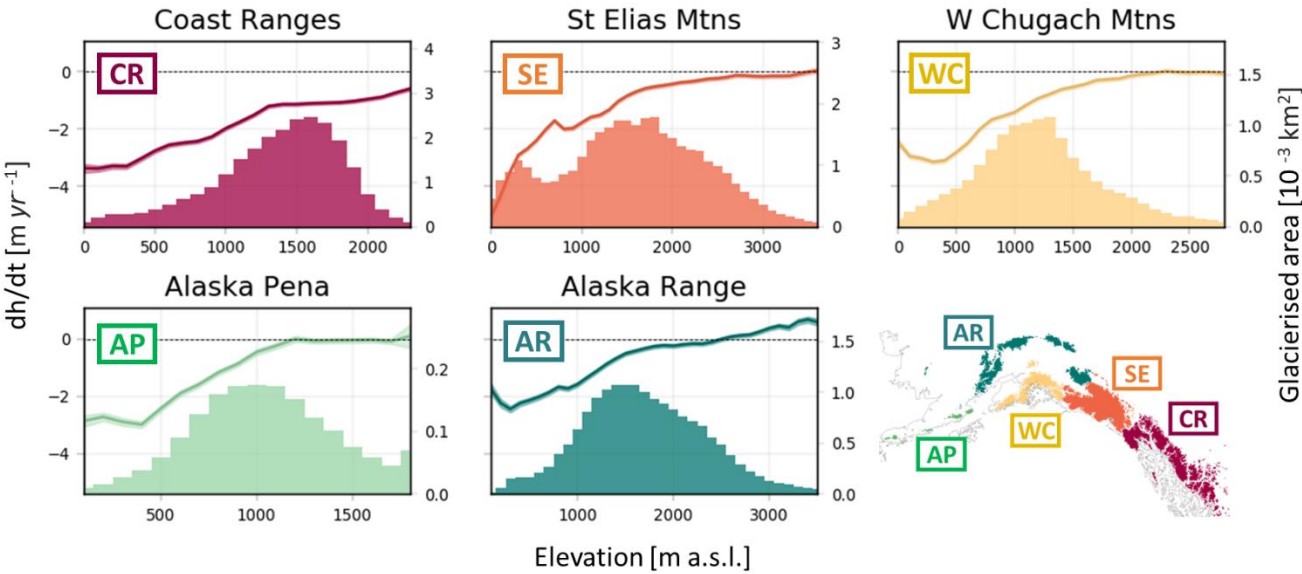

**Figure 11: Altitudinal distribution of elevation changes and glacier hypsometry functions in the Gulf of Alaska (GoA) in RGI 6.0 sub-regions between 2010 and 2019. The lines show elevation change rates with uncertainty envelopes plotted against 100 metre elevation bands (*left y-axis*). The bars display the glacier hypsometry (*right y-axis*).**

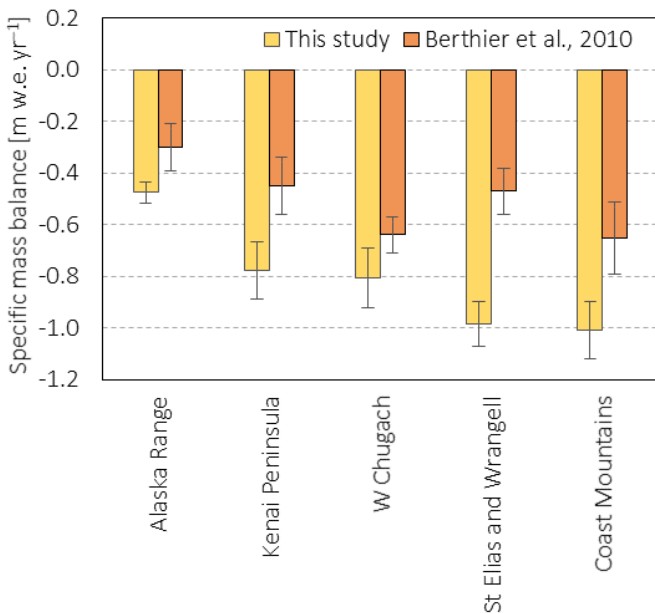

**Figure 12: Gulf of Alaska (GoA) specific mass balance trends, aggregated on the sub-regions by Berthier et al. (2010). The figure compares our results (2010 to 2019) to the estimates of Berthier et al. (2010), covering the time period 1962 to 2006.**

| | Glacier area [km²] | Specific mass change [m w.e. yr⁻¹] | Mass change [Gt yr⁻¹] |
|---|---|---|---|
| **W Tien Shan** | 9531 | −0.36 ± 0.07 | −3.42 ± 0.69 |
| **E Tien Shan** | 2854 | −0.47 ± 0.13 | −1.34 ± 0.37 |
| **C Himalaya** | 5447 | −0.43 ± 0.14 | −2.33 ± 0.75 |
| **W Kun Lun** | 8153 | +0.06 ± 0.05 | +0.51 ± 0.37 |
| **E Himalaya** | 4904 | −0.56 ± 0.16 | −2.76 ± 0.77 |
| **E Kun Lun** | 3251 | −0.47 ± 0.10 | −1.53 ± 0.32 |
| **Hengduan Shan** | 4383 | −0.98 ± 0.22 | −4.30 ± 0.98 |
| **Qilian Shan** | 1637 | −0.29 ± 0.22 | −0.47 ± 0.37 |
| **Inner Tibet** | 7923 | −0.26 ± 0.10 | −2.09 ± 0.80 |
| **S and E Tibet** | 3873 | −0.88 ± 0.32 | −3.38 ± 1.21 |
| **Hindu Kush** | 2938 | −0.27 ± 0.12 | −0.79 ± 0.35 |
| **Karakoram** | 22862 | −0.07 ± 0.02 | −1.49 ± 0.56 |

| | | | |
|---|---|---|---|
| **W Himalaya** | 7768 | −0.25 ± 0.09 | −1.94 ± 0.73 |
| **Hissar Alay** | 1846 | −0.21 ± 0.18 | −0.39 ± 0.33 |
| **Pamir** | 10234 | −0.23 ± 0.05 | −2.33 ± 0.54 |
| **Total** | 97604 | | |

**Table 1: High Mountain Asia (HMA) mass balance trends from 2010 to 2019, aggregated on the Randolph Glacier Inventory (RGI 6.0) sub- regions.**

| | Glacier area [km²] | Specific mass change [m w.e. yr⁻¹] | Mass change [Gt yr⁻¹] |
|---|---|---|---|
| **Alaska Range (Wrangell/Kilbuck)** | 16278 | −0.41 ± 0.05 | −6.6 ± 0.9 |
| **Alaska Pena (Aleutians)** | 1912 | −0.64 ± 0.10 | −1.2 ± 0.2 |
| **West Chugach Mountains (Talkeetna)** | 12052 | −0.80 ± 0.09 | −9.6 ± 1.0 |
| **Saint Elias Mountains** | 33174 | −1.03 ± 0.10 | −34.1 ± 3.4 |
| **Northern Coast Ranges** | 22963 | −1.08 ± 0.09 | −24.8 ± 2.1 |
| **Total** | 86379 | | |

**Table 2: Gulf of Alaska (GoA) mass balance trends from 2010 to 2019, aggregated on the Randolph Glacier Inventory (RGI 6.0) sub- regions.**