# Peer review of "Spatially and temporally resolved ice loss in High Mountain Asia and the Gulf of Alaska observed by CryoSat-2 swath altimetry between 2010 and 2019"

_The Cryosphere, 2020_

## Referee Comment (RC1) · Anonymous Referee #1 · 14 Sep 2020

Review of 'Ice loss in High Mountain Asia and the Gulf of Alaska observed by CryoSat-2 swath altimetry between 2010 and 2019' by L. Jakob et al.

Summary:

In this paper the authors have used a recent but well-established technique to quantify the glacial ice loss in two relatively large regions; 'High Mountain Asia' (HMA), including the Himalayas, and the Gulf of Alaska (GoA). The technique uses 'swath processing' of the Doppler beam-sharpened and interferometric altimeter data from CryoSat to measure the ice loss between commissioning (fall 2010) and the end of 2019.

General Comments:

Both areas include small and large mountain glaciers that are particularly challenging for any form of radar altimetry, and the authors are to be commended for both attempting this job and for the credible results they are presenting. Measuring recent mass loss in the HMA area and relating that to the changing weather conditions over the last nine years is important as so many people depend on and are affected by summer run-off.

Also, I think the authors can be commended for the comparison of their results with those of others who have used different techniques to estimate the change in mass balance of the glaciers in these regions. Figure 8 highlights the fact that some of the different approaches produce different results and, when the associated error estimates are considered, there are inconsistencies. I do not expect the authors to reconcile these problems, but I do want them to avoid the trap that some authors must have fallen into, i.e. underestimating their errors. While the random noise component can be estimated often there are other potential bias errors which can creep into the results. As these are hard to quantify, they are often dismissed or ignored. More on this in the detailed comments below.

For some of the areas the modulation in seasonal height change appears to be related to the magnitude of the snow accumulation as well as the summer melt. This is a consequence of the large number of height estimates possible with swath-processing and the temporal sampling in any one area. However, there is little discussion of this apparent capability in the paper, I think this is another area that could be explored further in this paper.

Specific Comments:

L9. '… *the largest non-steric contributor*'. I do not think the term 'non-steric' is appropriate here. If you mean the largest contributor after ocean thermal expansion, then use this more straight forward wording.

L25. I think reference to Bamber et al., 2018. (Env. Res. Letters) is more appropriate here than the Shepherd et al., 2020 reference. The later focusses on Greenland, not the non-ice sheet glaciers and ice caps while the Bamber reference includes discussion of change in both the large ice sheets, glaciers and ice caps.

L34. This paper will have wide readership and I think some of the technical terms should be explained, the term 'geodetic remote sensing methods' is used here and could (should?) be explained.

L39-L41. ICESat-2 (launched 2018) data is being, and will be, used to monitor change in the height of glacial ice worldwide. I think the combination of data from CryoSat-2 (CS-2) and ICESat-2 will lead to a better system than CryoSat alone. (ref. Smith et al. Science 2020 for ICESat-2 monitoring of glacial ice).

L100-L118. In the introduction the two limiting factors for the use of satellite radar altimeters over mountain glaciers are itemized… namely for CryoSat the limited 240 m range window and the closed loop on-board tracking used to position the start of the range window in fast time. But in the Data and Methods section there is no explicit explanation as to how these limitations are addressed or overcome with the methodology used in tis work.

L104-L106. The two sentences… '*SIRAL is a beam-forming… ground. The sensor emits… the beam.*' either need more explanation or could be left out completely. The approximate diameter of the area beneath the satellite from which returns might be expected is ~15 km, and the diameter of the first return footprint (POCA) from a flat Earth is ~ 1.5 km based on just the pulse bandwidth. It would be very rare indeed that a flat Earth model could be used in the areas studied here.

L112. '*Single*' should be 'signal', or 'return' power.

L122. I suggest… 'The distribution of height measurements departs…'

L123. The sentence 'Given the… domain' is poorly worded, try to simplify.

L125. In correcting for the difference between the glacier hypsometry in a 100 x 100 km bin and the height distribution of CS-2 swath measurements your methodology appears to discard legitimate measurements in those elevation bands which are over-populated in relation to the glacier hypsometry (Fig. S4). In summing the height change to get volume change why not simply scale the CS-2 results in the various elevation bands to match the hypsometry. Would this not be simpler and avoid discarding results?

L157. I am not sure that you can claim the temporal variation in CS-2 height change will always match the surface height change in either area. Looking at your seasonal CS-2 height change curves (Fig. 5) for some of the HMA there appears to be significant winter snow accumulation (5 – 10 m in some of your areas!) so I would suspect that conditions (the nature and density of the surface snow, and therefore reflectivity) will change between the winter and after the summer melt, and the 'penetration' or the effective surface seen by the radar altimeter will change with season and possibly elevation. For example, recent work on the high accumulation region in SE Greenland appears to show a seasonal change in the bias between the surface and CS-2 detected 'height' (Gray et al., Front. Earth Sci., https://doi.org/10.3389/feart.2019.00146). However, if you assume that conditions do not change significantly fall-to-fall then the year-to-year volume and mass change can still be estimated. The shape of the seasonal height change may be affected by a varying bias between the surface and the detected CS height. While I think your results are credible, I suspect that a demonstration that CS-2 seasonal height change matches surface height change in these areas will require a comparison of coincident CS-2 and ICESat-2 results.

L167. I would explain the term 'endorheic' or use the phrase 'closed or endorheic basins'.

L183. I am not sure how the percentage coverage of the glaciated regions in the two areas was estimated; the swath processed CS footprint area is ~ 380 m along-track times a figure dependent on

the cross-track slope and the waveform smoothing. For example, if the cross-track slope is 0.5 degrees then the footprint in this direction without smoothing is only 27 m. With waveform smoothing and volume backscatter this will be broadened, but it is still a fraction of the typical POCA footprint for flat terrain. Considering the early CryoSat results by Amaury Dehecq over the Himalayas, I have to say that the percentage of the glaciated areas you have covered (~ 50%?) is remarkable. Is this correct?

L185. Spatial coverage and elevation sampling… The relatively poor coverage of the lower reaches of the HMA and GoA glaciers is a concern. These are the normally the areas that are most vulnerable to rising temperatures and which often change the most.

L187. 'Whilst'

L192. *'Spatial coverage and number of points show a different relationship with hypsometry, which is due to the overlap between adjacent CryoSat footprints.'…* I don't understand this sentence, in the along-track direction the footprint is ~ 380 m and the sampling is ~ 300 m while in the cross-track direction there is oversampling if all the waveform points are retained after the waveform smoothing stage. In fact, if the filter smooths over 3 samples (from the Gourmelen reference) then you could use every third sample in the waveform.

L197. You have acknowledged the problem associated with the onboard tracking and the limited 240 m waveform window but the fact that most (?) of the glaciers you are studying will have termini at an elevation beyond the end of the measured waveform remains perhaps the most important limitation of your study. The implication that the small difference between the 'biased' and 'non-biased' estimates of the specific mass losses somehow justifies your results for mass loss is weak. A 'stable result' is not necessarily a precise one.

L209. '*In contrast with other studies (Brun et al., 2017; Shean et al., 2020) we find a heterogenous pattern in the Tibetan Plateau and Eastern Kunlun, with some scattered glaciers displaying higher mass losses.'…* Can you think why this would be the case?

L223. 'Temporal variability'. Looking at Figures S1, S2 and 5, I see the upward trend in elevation change after ~ 2015 for the Karakorum region but I am not so sure about the statement … '*This shift of thinning rates post-2015 is also clearly seen in Bhutan/East Himalaya, Kunlun (West and East), Tien Shan, Pamir Alay/Hissar Alay and Nyainqêntanglha/Hengduan 230 Shan (Figure 5, S1, S2)'.* What is clear is that the seasonal modulation is increasing for many of these areas. I admit that I have not studied glaciologic change in these areas previously, but can you have a 10 m height change for the glaciers of Pamir Allay between the summer of 2017 and the following winter? Even the 'Full region' plot, top left Figs. 5 and S2, shows the increasing seasonal modulation as well as the slow height decrease. Is this significant?

Looking at the elevation change rates in the HMA (Fig. S6) it seems that your study (Fig. S6a) produces a noisier elevation change rate vs normalized elevation than the Brun et al. study (Fig. S6b). Is this assessment fair? And can you rationalize this behavior?

Also, in Fig. 6, the variation in height change rate and the 100 m bin glacier hypsometry are plotted against elevation for the various regions. The elevation change-rates generally become less negative with increasing elevation but some of the curves are quite noisy and have what I would consider as suspicious jumps that are larger than the uncertainty shading. Can you provide some explanation for these variations?

L250 and onward… comparison of mass balance estimates. I will leave detailed comments on the difficulty of reconciling the various mass balance studies to others with a better knowledge of these geographic areas. The problem as I see it is that some of the studies appear to underestimate their potential errors. For the swath processing approach used in this study there are several potential issues which may lead to bias errors. While I acknowledge the careful approach used to try to eliminate the poor height values, I think the quoted errors may still not reflect all the potential problems that could lead to bias errors… For example:

1. As the orbit is essentially north-south for the HMA, is there any possible bias between the height change results dependent on glacier orientation (NS vs EW)? Along-track slopes (> ~ 1°) are not good news for the delay-Doppler or subsequent algorithms.
2. The swath-processed footprint is rarely contiguous, hopefully one area (on the glacier!) dominates the returns so that the phase can be used to geocode the footprint. Remember that with delay-Doppler processing the geocoding is done based on the differential phase which will be corrupted when multiple areas contribute to the range sample. The hope is that one area of the composite footprint dominates the return.
3. The seasonal variation in height change can reflect changing surface conditions as well as accumulation/melt. You cannot assume that the conditions in your areas are comparable to those studied in the papers you reference.
4. With swath-processing, compensation for the low percentage of results from the lower glacier elevations must be difficult. Looking at, e.g. Fig 11, the low elevation height change rate is quite variable, some of the areas show an increase in height loss with elevation when surely one would expect smoothly increasing loss with decreasing elevation?

In summary, while these results are on the one hand both impressive, important and well-illustrated, it is important that all the possible errors and biases are at least acknowledged. The 'uncertainty envelopes' used in the figures reflect the quantifiable errors in the methodology. While some of the potential bias errors are very difficult to quantify that does not excuse ignoring them. I would like the authors to at least acknowledge that there are other potential bias errors that could expand the 'uncertainty envelopes'.

---

## Referee Comment (RC2) · Tobias Bolch (Referee) · 30 Oct 2020

Review on the manuscript entitled "Ice loss in High Mountain Asia and the Gulf of Alaska observed by CryoSat-2 swath altimetry between 2010 and 2019" by L. Jacob et al. submitted to The Cryosphere

General remarks: The study provides estimations of glacier mass balance of two large glacierized regions: High Mountain Asia and the Gulf of Alaska using CryoSat-2 data. Most important is that the authors show the suitability of the radar altimeter to obtain not only information about one period but to obtain information about seasonal height and mass changes for the period 2010-2019. The method is not entirely novel but for

the first time applied over such large areas including smaller mountain glaciers. This is a timely and very relevant work as detailed information about glacier mass changes are important in many aspects. Overall, the manuscript is well written and illustrated. I am not a full expert in processing CryoSat-2 data, but as far as I can judge, the method applied seems sound and also the specific conditions of mountain glaciers are considered to degree possible. However, considering the nature of CryoSat-2 and judging the presented results I am not fully convinced that all results are reliable. As also detailed by the other reviewer there are many error sources and sources of uncertainties which should be better considered. I highlight below first the more general comments and provide then more detailed ones.

General comments:

1. Error/Uncertainty sources:

a. The authors subtract the TanDEM-X 90m DEM from Cryosat-2 swath elevation measurements. The state in their manuscript: "The remaining elevation differences are due to time-dependent elevation change that can be related to glacier thickness change as well as errors in the two data sets, temporal heterogeneity and differences in penetration between the reference DEM and the swath elevation measurements." Both utilised data are microwave data. Although the KU and X-band penetration is lower than the penetration of larger wavelengths such as the often used SRTM-C band data, it is not negligible especially in dry snow which is common in many parts of HMA. Moreover, the TanDEM-X DEM is composed of different acquisitions of different seasons and years.

b. Density conversion and snow accumulation: The author's apply a constant value of 850 kg/m$^3$. This value is often applied also in other studies but needs to be applied with caution. First of all Huss (2013) states that the conversion factor can be significant different for short periods. This is especially important as height changes of snow which has a much lower density can be large. Hence, the authors needs to consider

these density variation more carefully especially when interpreting short term changes.

c. These and other sources of uncertainties needs to be better acknowledged. I have the feeling that the provided uncertainty ranges of 0.03 and 0.07 m w.e./a are clearly too low. I suggest showing the accuracy of the data and results at few selected test sites with independent data (e.g. the HMA DEM, Shean et al. 2017, ICESat-2 data, detailed comparison to high resolution DEMs or other studies for smaller regions and also in-situ measurements, e.g. as available from WGMS).

2. Small glaciers and data coverage: The authors state that CryoSat-2 are also able to survey (very) small glaciers, but do not clearly state a size threshold what they consider as small. There are many very small glaciers especially in HMA which can have significant impact on the overall and especially specific mass changes. Please define "small" and show the portion of the size classes covered in comparison the glacier inventory. Moreover, be more specific about the data coverage and the representativeness and show a plot of the data coverage in relation the total area.

3. There is no mention of impact of glacier surges and avalanche-fed glaciers which are common in parts of the study regions. The validity of the data for these glacier types should be in an ideal case shown, but at least discussed.

4. The authors exclude the endorheic basis when considering the contribution to sea level rise. Here, the authors need to be more specific: Basins where part of the glacier melt as led to lake level rise (e.g. Neckel et al., 2014) this is suitable, but for the others the water (if not sored in the ground) would end up in the hydrological cycle and ultimately in the oceans.

5. Climatic consideration: The authors explain some of the variation by accumulation type and changes in weather and climate. While I agree that this is in principle true the relation to the weather and climate is too simplified. E.g. there are regions in the Tien Shan which receive more accumulation during summer and winter snowfall is also of high importance for parts central Himalaya. Please be here more specific. I suggest to

consider more references (e.g. Maussion et al. 2014) and consider Sakai and Fujita (2017) more in detail.

6. Sections on main results: Sections 3.2 and 3.3: are important sections as the main results are shown. However, these are really short and lack details. You have much more to show. Highlight here all the important results including the shorter term trends and seasonal variability.

7. Discussion sections: Put more emphasis on possible reasons of mismatches to other studies.

8. HIMAP-regions: In order to be able to better compare the regional results to other studies, I suggest to include also a comparison the subregions presented by the cryosphere chapter of the HIMAP report (Bolch et al. 2019) at least in a figure and table in the supplement. These regions were defined by an international team including local scientists and are often used (e.g. Shean et al. 2020, Rounce et al., 2020). Moreover, these regions capture better the regional variability (e.g. mass balances in East Pamir which are more positive and those in central and west Pamir which are more negative). However, I do not want to force you as I am one of the lead authors of this HIMAP chapter and leave this decision to you/the editor.

Specific comments:

Title: The title does not fully reflect the content. One of the strengths of the study is that is shows not only one period but annual and the seasonal variability.

Abstract: It is good to keep the abstract short. However, it would benefit if the variations of glacier mass/elevation change found are better highlighted (also in quantitative way).

L23: Write consistently "Glaciers and ice caps" and include also the percentage area of the glaciers as the area matters more when considering ice melt.

L26: I suggest to cite Immerzeel et al. (2020) instead of (2010).

L40: Include here also Shean et al. (2020)

L43-45: I suggest to include lines 25f here as it is a repetition apart from the seasonality. And please include also a suitable reference for food security and GLOFs.

L65ff: I suggest to introduce the subheading "Study regions" here.

L66: write "... includes the Himalaya, Tibetan mountain ranges, the Pamir and Tien Shan" (Shan=Mountain).

L67: write "about 95,500 (or 96,000 or even "almost 100,000") glaciers". Glacier number is a bit arbitrary and depends on the size threshold used and how you split contiguous ice masses.

L79: not only since the satellite records but also before. You may then cite here Bolch et al. (2012) which summarised the info for the Himalaya. Please also consider a reference for the Tien Shan which was not covered by the HIMAP report.

L83: See my comment on the glacier number above

L97: See my comment on L79. This applies also here.

L167: See my comment above. Be more specific.

L188: Omit "very". I would not anymore consider ASTER as "very-high resolution".

L206: The Karakoram anomaly was first introduced by Hewitt (2005) and then confirmed by geodetic measurements by Gardelle et al. (2012). Please cite these two references here.

L207: I am surprised to read about the moderate thinning for Lahaul-Spiti. In line with mass balance measurements and modelling of Chhota Shigri glacier (e.g. Azam et al. 2014), Mukherjee et al. (2018), showed significant mass loss in this region using geodetic data. However maybe mass loss was less after 2010? Please be more specific and quantitative and discuss in the discussion section.

L210ff: Be more specific about the own results here (and double check your results especially considering the uncertainty) and move the critical comparison to other studies to the discussion.

L213ff: Similar in this section: Be more specific about the own results and move the critical comparison to other studies to the discussion.

L224ff.: This section contains several interesting findings which I suggest presenting in the results section (e.g. the variability in mass changes for the different regions) and keep here, but extend the climatic discussion.

L249: This is again an important finding and should be move to the results section and the reasoning discussed here.

L264: Please avoid the term doubling or almost doubling. There is a clear increase of mass loss, but uncertainty ranges in this studies are (realistically calculated) and large. Consider here also the Study by King et al. (2019) who used similar data for the Himalaya but found less increase.

L296: "Widely discussed and predicted..." please provide evidence for that.

L385: Avoid citations in the conclusions (especially when too prominently referring to own work). Move this to the discussion for more details but keep the main statement here.

L305ff: I am not too familiar with all the work from Alaska, but I ask you to be more specific regarding the work you are considering. Some have clear different time periods of analysis. As mentioned above discuss in more detail possible reasons to different results.

Some remarks on figure 2: The overall pattern of mass changes makes sense and fits to the current knowledge. There are, however, certain 100x100 grids where the mass balance does not fit. This is especially the case for the central part of Northern Tien Shan in Kazakhstan. There was a positive balance according to your data,

but both geodetic data and in-situ measurements of Tuyuksu glacier clearly highlight negative mass balance Kapitsa et al. (2020) and WGMS data. As already mentioned above Lahaul-Spiti has also more negative values in other studies. You may compare to Chhota Shigri glacier which was shown to be representative for the region. In contrast Eastern Pamir and Altun Shan seems more negative than suggested by other studies. This highlights again that a more careful uncertainty analysis and comparison to existing data and studies are needed.

References cited in my review but not in the manuscript:

Azam, M.F., Wagnon, P., Vincent, C., Ramanathan, A., Linda, A., Singh, V.B., 2014. Reconstruction of the annual mass balance of Chhota Shigri glacier, Western Himalaya, India, since 1969. Ann. Glaciol. 55 (66), 69–80. https://doi.org/10.3189/2014AoG66A104.

Hewitt, K., 2005. The Karakoram Anomaly? Glacier expansion and the "Elevation Effect" Karakoram Himalaya. Mount. Res. Dev. 25 (4), 332–340.

Immerzeel et al., 2020. Importance and vulnerability of the world's water towers. Nature 577 (7790), 364–369. https://doi.org/10.1038/s41586-019-1822-y.

Kapitsa, V., Shahgedanova, M., Severskiy, I., Kasatkin, N., White, K., Usmanova, Z., 2020. Assessment of Changes in Mass Balance of the Tuyuksu Group of Glaciers, Northern Tien Shan, Between 1958 and 2016 Using Ground-Based Observations and Pléiades Satellite Imagery. Frontiers in Earth Science 8, 259. https://doi.org/10.3389/feart.2020.00259.

King, O., Bhattacharya, A., Bhambri, R., Bolch, T., 2019. Glacial lakes exacerbate Himalayan glacier mass loss. Scientific Reports 9 (1), 18145. https://doi.org/10.1038/s41598-019-53733-x.

Maussion, F., Scherer, D., Mölg, T., Collier, E., Curio, J., Finkelnburg, R., 2014. Precipitation seasonality and variability over the Tibetan Plateau as resolved by the High Asia

[Figure]

Reanalysis. J. Climate 27, 1910–1927. https://doi.org/10.1175/JCLI-D-13-00282.1.

Mukherjee, K., Bhattacharya, A., Pieczonka, T., Ghosh, S., Bolch, T., 2018. Glacier mass budget and climate reanalysis data indicate a climatic shift around 2000 in Lahaul-Spiti, western Himalaya. Climatic Change 148 (1), 219–233. https://doi.org/10.1007/s10584-018-2185-3.

Shean, D.E., Bhushan, S., Montesano, P., Rounce, D.R., Arendt, A., Osmanoglu, B., 2020. A systematic, regional assessment of High Mountain Asia glacier mass balance. Frontiers in Earth Science 7, 363. https://doi.org/10.3389/feart.2019.00363.

Shean, D.E., 2017. High Mountain Asia 8-meter DEM mosaics derived from optical imagery, version 1. https://doi.org/10.5067/KXOVQ9L172S2.

---

## Author Comment (AC1) · 14 Dec 2020

Response to comments of reviewer 1 to:

**Spatially and temporally resolved ice loss in High Mountain Asia and the Gulf of Alaska observed by CryoSat-2 swath altimetry between 2010 and 2019**

In this paper the authors have used a recent but well-established technique to quantify the glacial ice loss in two relatively large regions; 'High Mountain Asia' (HMA), including the Himalayas, and the Gulf of Alaska (GoA). The technique uses 'swath processing' of the Doppler beam-sharpened and interferometric altimeter data from CryoSat to measure the ice loss between commissioning (fall 2010) and the end of 2019.

**Main comments:**

Both areas include small and large mountain glaciers that are particularly challenging for any form of radar altimetry, and the authors are to be commended for both attempting this job and for the credible results they are presenting. Measuring recent mass loss in the HMA area and relating that to the changing weather conditions over the last nine years is important as so many people depend on and are affected by summer run-off.

Also, I think the authors can be commended for the comparison of their results with those of others who have used different techniques to estimate the change in mass balance of the glaciers in these regions. Figure 8 highlights the fact that some of the different approaches produce different results and, when the associated error estimates are considered, there are inconsistencies. I do not expect the authors to reconcile these problems, but I do want them to avoid the trap that some authors must have fallen into, i.e. underestimating their errors. While the random noise component can be estimated often there are other potential bias errors which can creep into the results. As these are hard to quantify, they are often dismissed or ignored. More on this in the detailed comments below

Answer: This will be answered in the specific comments below.

For some of the areas the modulation in seasonal height change appears to be related to the magnitude of the snow accumulation as well as the summer melt. This is a consequence of the large number of height estimates possible with swath-processing and the temporal sampling in any one area. However, there is little discussion of this apparent capability in the paper, I think this is another area that could be explored further in this paper.

Answer: Thank you for this valuable input. We have expanded this aspect in the following sections (for more details refer to the specific comments below):

Time series of surface elevation changes:

"Although time-series are generally reflecting the actual change in surface elevation, there are a number of limitations that are important to keep in mind when interpreting the results from radar altimetry. For the reasons stated above, scattering properties can induce elevation biases at seasonal time-scale (Gray et al., 2019). In addition, integrating changes over very large regions can lead to spatial heterogeneity in the successive time steps, in particular when the data volume becomes too low. These limitations may explain some of the observed patterns, and in particular the few cases where seasonal variability is larger than what is expected from our knowledge of SMB in the regions."

Mass balance and contribution to sea level rise:
"To obtain volume changes we use the glacierised area of the Randolph Glacier Inventory (RGI 6.0) (RGI Consortium, 2017). We assume the standard bulk density of 850 kg/m$^3$ (Huss, 2013) to convert volume changes to equivalent mass changes. This assumption is considered appropriate for a wide range of conditions and longer-term trends, however, this factor can differ significantly for shorter term periods (<3 years) (Huss, 2013)."

Conclusion:
"This is the first study to demonstrate the ability of interferometric radar altimetry to monitor large-scale change in thickness, mass and sea-level contribution of glaciers across regions of extreme topography. This, along with recent work in the Arctic and Patagonia demonstrates the potential of such a system to monitor trends in ice mass on a global scale and with increased temporal resolution. **It also demonstrates the ability to monitor monthly change and paves the way to an observation-based quantification of seasonal accumulation and melting processes, a task that will likely require combination with regional climate models, and with other sensors such as IceSat-2 and high-resolution DEMs.**"

**Main comments:**

L9. '… the largest non-steric contributor'. I do not think the term 'non-steric' is appropriate here. If you mean the largest contributor after ocean thermal expansion, then use this more straight forward wording.

Answer: We have rephrased accordingly:
"Glaciers are currently **the largest contributor to sea level rise after ocean thermal expansion**, contributing ~30% to sea level budget."

L25. I think reference to Bamber et al., 2018. (Env. Res. Letters) is more appropriate here than the Shepherd et al., 2020 reference. The later focusses on Greenland, not the non-ice sheet glaciers and ice caps while the Bamber reference includes discussion of change in both the large ice sheets, glaciers and ice caps.

Answer: We cited Shepherd et al. (2020) to highlight that glaciers (Gardner et al., 2013; Wouters et al., 2019; Zemp et al., 2019) contribute more than Greenland (Shepherd et al., 2020). We rephrase this section as follows:

"Glaciers store less than 1% of the mass (Farinotti et al., 2019) and occupy just over 4% of the area (RGI Consortium, 2017) of global land ice, however their rapid rate of mass loss accounts for almost a third of the global sea level rise, the largest sea level rise (SLR) contribution from land-ice (Bamber et al., 2018; Gardner et al., 2013; Slater et al., 2020; Wouters et al., 2019; Zemp et al., 2019)."

L34. This paper will have wide readership and I think some of the technical terms should be explained, the term 'geodetic remote sensing methods' is used here and could (should?) be explained.

Answer: We revised accordingly:
"The traditional approach (glaciological method) extrapolates in situ observations (Bolch et al., 2012; Cogley, 2011; Yao et al., 2012; Zemp et al., 2019), however measurements are sparse and possibly biased towards better accessible glaciers located at lower altitudes (Fujita and Nuimura, 2011; Gardner et al., 2013; Wagnon et al., 2013). **In contrast, geodetic remote sensing methods rely on comparisons of topographic data or gravity fields to determine glacier changes. Recent geodetic remote sensing methods include** (1) Digital Elevation Model (DEM) differencing (Berthier et al., 2010; Brun et al., 2017; Gardelle et al., 2013; Maurer et al., 2019; Shean et al., 2020), (2) satellite laser altimetry (Kääb et al., 2012, 2015; Neckel et al., 2014; Treichler et al., 2019) and (3) Gravity Recovery and Climate experiment (GRACE) satellite gravimetry (Ciracì et al., 2020; Gardner et al., 2013; Jacob et al., 2012; Luthcke et al., 2008; Wouters et al., 2019)."

L39-L41. ICESat-2 (launched 2018) data is being, and will be, used to monitor change in the height of glacial ice worldwide. I think the combination of data from CryoSat-2 (CS-2) and ICESat-2 will lead to a better system than CryoSat alone. (ref. Smith et al. Science 2020 for ICESat-2 monitoring of glacial ice).

Answer: We fully agree with the reviewer.

In the conclusion section:
"This, along with recent work in the Arctic and Patagonia demonstrates the potential of such a system to monitor trends in ice mass on a global scale and with increased temporal resolution. **It also demonstrates the ability to monitor monthly change and paves the way to an observation-based quantification of seasonal accumulation and melting processes, a task that will likely require combination with regional climate models, and with other sensors such as IceSat-2 and high-resolution DEMS."**

L100-L118. In the introduction the two limiting factors for the use of satellite radar altimeters over mountain glaciers are itemized… namely for CryoSat the limited 240 m range window and the closed loop on-board tracking used to position the start of the range window in fast time. But in the Data and Methods section there is no explicit explanation as to how these limitations are addressed or overcome with the methodology used in this work.

Answer:

The short answer is that they are not. If the onboard tracking positions the range window outside the range of elevation at which glaciers are found by more than ~half the range window size, there will be no data to exploit over glaciers in any of the recorded echoes. However the wording of this section in the introduction is ambiguous, in addition, Swath will help sample a larger range of elevation within the 240m range window and so we edited the text as follows:

In the introduction:
"**Over regions of more extreme surface topography however, such as those found in mountain glacier areas, the use of radar altimetry has been prohibited by the large pulse-limited footprint, a limited range window (240 m for CryoSat), and closed-loop onboard tracking used to position the altimeter's range window (Dehecq et al., 2013). Despite these limitations, CryoSat's sharper footprint and interferometric capabilities have led to promising studies over mountain glaciers (Dehecq et al., 2013; Foresta et al., 2018; Trantow and Herzfeld, 2016)**"

In the data and method:
" In contrast, swath altimetry exploits the full radar waveform to map a dense swath (~5 km wide) of elevation measurements across the satellite ground track beyond POCA (Foresta et al., 2016, 2018; Gourmelen et al., 2018; Gray et al., 2013; Hawley et al., 2009) providing one to two orders of magnitude more elevation measurements compared with POCA **and improving the sampling of topographic lows (Foresta at al., 2016)**. This makes the CryoSat-2 sensor at present the only radar altimeter able to survey small glaciers at high resolution."

L104-L106. The two sentences… '*SIRAL is a beam-forming… ground. The sensor emits… the beam.*' either need more explanation or could be left out completely. The approximate diameter of the area beneath the satellite from which returns might be expected is ~15 km, and the diameter of the first return footprint (POCA) from a flat Earth is ~ 1.5 km based on just the pulse bandwidth. It would be very rare indeed that a flat Earth model could be used in the areas studied here.

Answer: We revised accordingly:

"SIRAL is a beam-forming active microwave radar altimeter with a maximum imaging range of ~15 km on the ground. The sensor emits time-limited Ku-band pulses aimed at reducing the footprint to ~1.6 km within the beam. Over land-ice, the sensor operates in synthetic aperture interferometric (SARIn) mode, which allows delay-Doppler processing to generate an along-track footprint of ~380 m, while cross-track interferometry is used to extract key information about the position of the footprint centre. In practice however, footprint size will vary depending on properties such as surface slopes, scattering properties, and distance from the Point-of-Closest-Approach (POCA)."

L112. 'Single' should be 'signal', or 'return' power.

Answer: We have changed this to "received power".

L122. I suggest… 'The distribution of height measurements departs…'

Answer: We have changed this to: "**The distribution of elevation measurements with altitude** departs somewhat from the glaciers' hypsometry".

L123. The sentence 'Given the… domain' is poorly worded, try to simplify.

Answer: We have revised accordingly:
"**Hypsometric representativeness of measurements within spatial units is a key requirement for robust glacier trend estimates. A bias in the altitudinal distribution of observations can lead to a bias in the total rate of thinning when integrated over a larger domain, as rate of thickness change is often strongly correlated with altitude**."

L125. In correcting for the difference between the glacier hypsometry in a 100 x 100 km bin and the height distribution of CS-2 swath measurements your methodology appears to discard legitimate measurements in those elevation bands which are over-populated in relation to the glacier hypsometry (Fig. S4). In summing the height change to get volume change why not simply scale the CS-2 results in the various elevation bands to match the hypsometry. Would this not be simpler and avoid discarding results?

Answer: We have initially tested the scaling approach too, and concluded that we achieve better results with the discard method we currently use. We found that the scaling approach can give high weighting to very few measurement points in sparse areas, which can lead to a reduction of the quality of results. It is also important to note that our current method rates the quality of each measurement and discards the lowest quality elevation measurements first, which can increase the stability of results in dense areas.

L157. I am not sure that you can claim the temporal variation in CS-2 height change will always match the surface height change in either area. Looking at your seasonal CS-2 height change curves (Fig. 5) for some of the HMA there appears to be significant winter snow accumulation (5 – 10 m in some of your areas!) so I would suspect that conditions (the nature and density of the surface snow, and therefore reflectivity) will change between the winter and after the summer melt, and the 'penetration' or the effective surface seen by the radar altimeter will change with season and possibly elevation. For example, recent work on the high accumulation region in SE Greenland appears to show a seasonal change in the bias between the surface and CS-2 detected 'height' (Gray et al., Front. Earth Sci., https://doi.org/10.3389/feart.2019.00146). However, if you assume that conditions do not change significantly fall-to-fall then the year-to-year volume and mass change can still be estimated. The shape of the seasonal height change may be affected by a varying bias between the surface and the detected CS height. While I think your results are credible, I suspect that a demonstration that CS-2 seasonal height change matches surface height change in these areas will require a comparison of coincident CS-2 and ICESat-2 results.

We agree with the reviewer, our wording was ambiguous. We rephrase as follow:

"It is a well known observation that microwave pulses scatter from the surface as well as the subsurface, which can lead to elevation change bias in regions of historically anomalous melt event (Nilsson et al., 2015); **or at seasonal time scale (Gray et al., 2019)**. Over most regions however, it has been shown that surface elevation change from CryoSat **over**

**annual and pluri-annual time scale are** consistent with in-situ, airborne, and meteorological observations (Gourmelen et al., 2018; Gray et al., 2015, **2019**; McMillan et al., 2014a; Zheng et al., 2018)"

L167. I would explain the term 'endorheic' or use the phrase 'closed or endorheic basins'.

Answer: We have changed 'endorheic' to 'closed or endorheic'

L183. I am not sure how the percentage coverage of the glaciated regions in the two areas was estimated; the swath processed CS footprint area is ~ 380 m along-track times a figure dependent on the cross-track slope and the waveform smoothing. For example, if the cross-track slope is 0.5 degrees then the footprint in this direction without smoothing is only 27 m. With waveform smoothing and volume backscatter this will be broadened, but it is still a fraction of the typical POCA footprint for flat terrain. Considering the early CryoSat results by Amaury Dehecq over the Himalayas, I have to say that the percentage of the glaciated areas you have covered (~ 50%?) is remarkable. Is this correct?

Answer: To calculate coverage, we assume an area the size of the pulse limited footprint at POCA for flat terrain i.e. ~380m by 1.5km. However the reviewer is correct that the real footprint will be dependent on many factors, several of which are not fully controlled. As the reviewer rightly points out, data filtering which will tend to coarsen across track resolution, surface slope will decrease across track footprint size, scattering process, e.g. a certain amount of volume scattering, will increase the footprint size for a given surface slope, the footprint size will also vary with distance from POCA. We have rephrased this section as follows:

"Using the **theoretical** pulse-limited footprint size of CryoSat-2, we **derive** a total spatial coverage of glaciated regions of 55% in the GoA and 32% in HMA respectively."

We cover 32% of the glaciated area, 50% is just an estimate of the potential assuming that onboard tracking was not an issue. We rephrase that sentence as follows:

"Given that it is estimated that 40% of HMA glaciers are not sampled due to onboard tracking limitations (Dehecq et al., 2013), we estimate that with an appropriate onboard tracking system, the rate coverage for HMA would be as high as 50%."

L185. Spatial coverage and elevation sampling… The relatively poor coverage of the lower reaches of the HMA and GoA glaciers is a concern. These are the normally the areas that are most vulnerable to rising temperatures and which often change the most.

Answer: A comparison of the glacier hypsometry and the spatial coverage of our data (Figure S4) shows that we still achieve good coverage at low elevations in both regions. In addition, we interpolate missing data based on the relationship between elevation and elevation changes. We therefore still capture the changes in the lower reaches of the HMA and GoA glaciers.

We have revised part of the spatial coverage and elevation sampling section accordingly:

"We observe a bias of the total number of swath measurements towards higher altitudes (e.g. Figure S5), which can be attributed to the onboard tracking tending to favour elevations closest to the satellite. However, a comparison of the glacier hypsometry and the spatial coverage of our data (Figure S4) shows that we still achieve good coverage at low elevations in both regions. In addition, we interpolate missing data based on the relationship between elevation and elevation changes and therefore still capture the changes in the lower reaches of the HMA and GoA glaciers."

[Figure]

**Figure S4: Figure (a) and (b) display glacier hypsometry (light blue) and swath elevation data coverage (grey) for the two regions High Mountain Asia (a) and the Gulf of Alaska (b). Figure (c) and (d) display data coverage as a percentage of the glacier hypsometry and figure (e) and (f) display the cumulative glacier area.**

L187. 'Whilst'

Answer: We have changed this.

L192. 'Spatial coverage and number of points show a different relationship with hypsometry, which is due to the overlap between adjacent CryoSat footprints.'… I don't understand this sentence, in the along-track direction the footprint is ~ 380 m and the sampling is ~ 300 m while in the cross-track direction there is oversampling if all the waveform points are retained after the waveform smoothing stage. In fact, if the filter smooths over 3 samples (from the Gourmelen reference) then you could use every third sample in the waveform.

Answer: We agree that in theory, successive samples should be independent, but for the reason given above e.g. in particular with respect to volume scattering, and our approach to modeling the coverage i.e. using true sample location and fixed footprint size, this leads to overlap in our modelled footprints.

We rephrased as follows:

**"Using the theoretical pulse-limited footprint size of CryoSat-2, we derive a total spatial coverage of glaciated regions of 55% in the GoA and 32% in HMA respectively. These values are the combined result of the absence of recorded returns due to orbit separation and onboard-tracking limitation (Dehecq et al., 2013), and of data quality. Given that it is estimated that 40% of HMA glaciers are not sampled due to onboard tracking limitations (Dehecq et al., 2013) we estimate that with an appropriate onboard tracking system, the rate coverage for HMA would be as high as 50%.** These values are within the high-end of the range of observational methods (Zemp et al., 2019), whilst generally lower than the coverage provided by high resolution sensors (Brun et al., 2017; Shean et al., 2020). As expected from the relatively large footprint of radar altimeters, we do observe a positive correlation between spatial coverage and glacier size, we do however observe coverage over all glacier sizes (Figure S6). **We observe a bias of the total number of swath measurements towards higher altitudes (e.g. Figure S5), which can be attributed to the onboard tracking tending to favour elevations closest to the satellite. However, a comparison of the glacier hypsometry and the spatial coverage of our data (Figure S4) shows that we still achieve good coverage at low elevations in both regions."**

L197. You have acknowledged the problem associated with the onboard tracking and the limited 240 m waveform window but the fact that most (?) of the glaciers you are studying will have termini at an elevation beyond the end of the measured waveform remains perhaps the most important limitation of your study. The implication that the small difference between the 'biased' and 'non-biased' estimates of the specific mass losses somehow justifies your results for mass loss is weak. A 'stable result' is not necessarily a precise one.

Answer: As discussed above, we have measurements covering all elevation ranges on glaciers in both regions, including glacier termini (Figure S4). Also refer to answer of L185.

We have revised part of the spatial coverage and elevation sampling section accordingly:

**"We observe a bias of the total number of swath measurements towards higher altitudes (e.g. Figure S5), which can be attributed to the onboard tracking tending to favour elevations closest to the satellite. However, a comparison of the glacier hypsometry and the spatial coverage of our data (Figure S4) shows that we still achieve good coverage at low elevations in both regions. In addition, we interpolate missing data based on the relationship between elevation and elevation changes and therefore still capture the changes in the lower reaches of the HMA and GoA glaciers."**

L209. 'In contrast with other studies (Brun et al., 2017; Shean et al., 2020) we find a heterogeneous pattern in the Tibetan Plateau and Eastern Kunlun, with some scattered glaciers displaying higher mass losses.'… Can you think why this would be the case?

We have made adaptations in the elevation change rate calculation, integrating the stability of our regression results to identify results that are particularly sensitive to data sampling, data distribution and data weighting. This method adaptation is described in more detail in the response to reviewer #2.
The updated results display a more homogeneous pattern in the Tibetan Plateau, Eastern Kunlun and in Nyainqêntanglha/Hengduan Shan, which is more comparable to other studies (Brun et al., 2017, Shean et al., 2020). We have therefore removed this statement.

L223. 'Temporal variability'. Looking at Figures S1, S2 and 5, I see the upward trend in elevation change after ~ 2015 for the Karakoram region but I am not so sure about the statement … 'This shift of thinning rates post-2015 is also clearly seen in Bhutan/East Himalaya, Kunlun (West and East), Tien Shan, Pamir Alay/Hissar Alay and Nyainqêntanglha/Hengduan 230 Shan (Figure 5, S1, S2)'. What is clear is that the seasonal modulation is increasing for many of these areas. I admit that I have not studied glaciologic change in these areas previously, but can you have a 10 m height change for the glaciers of Pamir Allay between the summer of 2017 and the following winter? Even the 'Full region' plot, top left Figs. 5 and S2, shows the increasing seasonal modulation as well as the slow height decrease. Is this significant?

Answer: The reviewer is correct that changes described above are subtle and seasonal variability relatively large. We therefore removed mention of these later regions. It is not clear to us why seasonal amplitude increases for some regions over time, it is not necessarily systematic however, neither at the scale of the HMA regions, nor when looking at the GoA region. As far as the overall HMA time-series is concerned, apart in 2018, the seasonality is relatively constant. We tend to observe a lower data density when these larger values occur and so these could be the result of a decrease in quality of spatial sampling during specific time-periods.

We have added content in the time-series sub-section of the method section to mention sources of uncertainty in the current time-series calculation:
**"Although time-series are generally reflecting the actual change in surface elevation, there are a number of limitations that are important to keep in mind when interpreting the results from radar altimetry. For the reasons stated above, scattering properties can induce elevation biases at seasonal time-scale (Gray et al., 2019). In addition, integrating changes over very large regions can lead to spatial heterogeneity in the**

**successive time steps, in particular when the data volume becomes too low. These limitations may explain some of the observed patterns, and in particular the few cases where seasonal variability is larger than what is expected from our knowledge of SMB in the regions."**

Looking at the elevation change rates in the HMA (Fig. S6) it seems that your study (Fig. S6a) produces a noisier elevation change rate vs normalized elevation than the Brun et al. study (Fig. S6b). Is this assessment fair? And can you rationalize this behavior?

Answer: The study Brun et al. (2017) – and other DEM differencing studies such as Shean et al. (2020) – resolve elevation changes at much higher spatial resolution than we are currently able to generate with CryoSat-2, leading to smoother curves. However, the plot demonstrates that despite the challenges we face with radar altimetry in the complex terrain of High Mountain Asia we are still able to resolve the relationship between hypsometry and elevation changes which is comparable to results of other studies.

Also, in Fig. 6, the variation in height change rate and the 100 m bin glacier hypsometry are plotted against elevation for the various regions. The elevation change-rates generally become less negative with increasing elevation but some of the curves are quite noisy and have what I would consider as suspicious jumps that are larger than the uncertainty shading. Can you provide some explanation for these variations?

Answer: We do observe a general correlation between uncertainty and variability of the hypsometric curves, although it is true that the magnitude of the envelope is often narrower - this touches upon other such comments in the review about uncertainty. What we do observe is that these regions of larger variability occur for sectors with lower glacier area. We make a note of this in the manuscript:

"**While some variability exists along the profiles, in particular over regions and elevation ranges containing fewer glaciers that can reflect a less robust solution and or spatial variability in glacier response, trends between elevation and ice thickness change are clearly visible.**"

L250 and onward… comparison of mass balance estimates. I will leave detailed comments on the difficulty of reconciling the various mass balance studies to others with a better knowledge of these geographic areas. The problem as I see it is that some of the studies appear to underestimate their potential errors. For the swath processing approach used in this study there are several potential issues which may lead to bias errors. While I acknowledge the careful approach used to try to eliminate the poor height values, I think the quoted errors may still not reflect all the potential problems that could lead to bias errors…

For example:
1. As the orbit is essentially north-south for the HMA, is there any possible bias between the height change results dependent on glacier orientation (NS vs EW)? Along-track slopes (> ~ 1°) are not good news for the delay-Doppler or subsequent algorithms.

2. The swath-processed footprint is rarely contiguous, hopefully one area (on the glacier!) dominates the returns so that the phase can be used to geocode the footprint. Remember

that with delayDoppler processing the geocoding is done based on the differential phase which will be corrupted when multiple areas contribute to the range sample. The hope is that one area of the composite footprint dominates the return.

3. The seasonal variation in height change can reflect changing surface conditions as well as accumulation/melt. You cannot assume that the conditions in your areas are comparable to those studied in the papers you reference.

4. With swath-processing, compensation for the low percentage of results from the lower glacier elevations must be difficult. Looking at, e.g. Fig 11, the low elevation height change rate is quite variable, some of the areas show an increase in height loss with elevation when surely one would expect smoothly increasing loss with decreasing elevation?

Answer:

We agree in general with the reviewer, we respect to the specific made:

1.The reviewer is correct about the importance of aspect and slopes. In the three graphs below we attempt to quantify the impact of both on data accuracy and distribution. The first plot (Figure S10) shows the comparison between swath elevation and the reference DEM. We do indeed observe an increase in the dispersion of the elevation difference with increasing along and across track slopes, while the bias remains relatively similar. Note that the dispersion will be accounted for in our current error method. The other impact of high along track slopes will be in the ability of the onboard tracker to keep track of the surface, with elevated along-track slopes the onboard tracker will be more likely to "lose lock" and this will be reflected in the data loss discussed in Dehecq et al., (2013) - this is a strong reason to develop a more suited onboard tracker for future radar altimetry missions. We added a mention of this in the manuscript, in the new error section discussed at the end of this specific response. We do not observe significant differences between the distribution of swath elevation versus aspect and the distribution of glacier aspect as seen in the last graph below (Figure S11).

[Figure]

**Figure S10: Differences between the TanDEM-X 90m DEM (German Aerospace Center [DLR], 2018) and the swath elevation measurements in a study area in High Mountain Asia (HMA) for different along-track and across-track slopes, including median average deviation (MAD) and mean (MEAN) of the elevation differences (referred to as *elevDiff*).**

[Figure]

**Figure S11: Distribution of aspect in our swath elevation measurements (a) and the distribution of glacier aspects (b) in a study region in High Mountain Asia (HMA).**

3. The reviewer is correct, we have made this clear now in the method sections related to rates of elevation change and in the time-series section.

4. Other studies of the Gulf of Alaska have reported a similar relationship to this study (e.g. refer to Berthier et al. (2010)), and attributed this to the impact of debris cover. In addition the use of the static RGI glacier masks can also impact the elevation change profile, by including areas in the profile where glaciers have retreated since the date of the masks.

We mention this in section 4.2.2:
"In the Western Chugach Mountains, Alaska Range and Alaska Peninsula we observe a decrease of thinning rates towards the lowest elevations of these sub-regions, which can be attributed to the effect of debris cover and the temporal evolution of glacier extent during the study period, one of the limitations when using static glacier masks. This characteristic has been observed, although more pronounced and across all sub-regions, by Berthier et al. (2010) and Arendt et al. (2002)."

To cover the various points made above, we added the following paragraph in the method section:

**"While our uncertainty methods follow existing approaches and our error bounds are similar in magnitude to Brun et al. (2017), Kääb et al. (2012) and Shean et al. (2020) but lower than GRACE-based estimates, several additional potential sources of errors could impact the results, and methods to assess them, not currently available, should be developed. Radar altimetry has been shown to be sensitive to surface slopes, and in particular to slope in the direction of the satellite's flight path, in regions like HMA and GoA this impact will also be seen in the performance of the onboard tracker as for large slopes the system is expected to "lose lock". It is a well-known observation that microwave pulses scatter from the surface as well as the subsurface, which can lead to elevation change bias in regions of historically anomalous melt events (Nilsson et al., 2015); or at seasonal time-scale (Gray et al., 2019). Over most regions however, it has been shown that surface elevation change from CryoSat over annual and pluri-annual time scale are consistent with in-situ, airborne, and meteorological observations (Gourmelen et al., 2018; Gray et al., 2015, 2019; McMillan et al., 2014a; Zheng et al., 2018). Using static glacier masks can also lead to errors in regions of rapid dynamic changes. In general these limitations are known and efforts are currently underway in the community to improve uncertainty analysis, and develop new glaciers outlines products."**

In summary, while these results are on the one hand both impressive, important and well-illustrated, it is important that all the possible errors and biases are at least acknowledged. The 'uncertainty envelopes' used in the figures reflect the quantifiable errors in the methodology. While some of the potential bias errors are very difficult to quantify that does not excuse ignoring them. I would like the authors to at least acknowledge that there are other potential bias errors that could expand the 'uncertainty envelopes'.

We appreciate the feedback and agree with the reviewer that more should be done as a community to understand, quantify, and reconcile the several estimates available today. Several community initiatives are underway. While the purpose of this study was not to resolve these issues, we hope that the additional context provided here and in the updated manuscript help make it clear that we do share the reviewer's thoughts.

---

## Author Comment (AC2) · 14 Dec 2020

Response to comments of reviewer 2 to:

**Spatially and temporally resolved ice loss in High Mountain Asia and the Gulf of Alaska observed by CryoSat-2 swath altimetry between 2010 and 2019**

The study provides estimations of glacier mass balance of two large glacierized regions: High Mountain Asia and the Gulf of Alaska using CryoSat-2 data. Most important is that the authors show the suitability of the radar altimeter to obtain not only information about one period but to obtain information about seasonal height and mass changes for the period 2010-2019. The method is not entirely novel but for the first time applied over such large areas including smaller mountain glaciers. This is a timely and very relevant work as detailed information about glacier mass changes are important in many aspects. Overall, the manuscript is well written and illustrated. I am not a full expert in processing CryoSat-2 data, but as far as I can judge, the method applied seems sound and also the specific conditions of mountain glaciers are considered to degree possible. However, considering the nature of CryoSat-2 and judging the presented results I am not fully convinced that all results are reliable. As also detailed by the other reviewer there are many error sources and sources of uncertainties which should be better considered. I highlight below first the more general comments and provide then more detailed ones.

**General comments:**

1. Error/Uncertainty sources:
a. The authors subtract the TanDEM-X 90m DEM from Cryosat-2 swath elevation measurements. The state in their manuscript: "The remaining elevation differences are due to time-dependent elevation change that can be related to glacier thickness change as well as errors in the two data sets, temporal heterogeneity and differences in penetration between the reference DEM and the swath elevation measurements." Both utilised data are microwave data. Although the KU and X-band penetration is lower than the penetration of larger wavelengths such as the often used SRTM-C band data, it is not negligible especially in dry snow which is common in many parts of HMA. Moreover, the TanDEM-X DEM is composed of different acquisitions of different seasons and years.

Answer: We agree with the reviewer and in effect this corresponds to our statement. We are considering these errors in our current uncertainty analysis, in that the difference in scattering depth and multi-annual composite of the reference DEM will result in a larger spread of the corrected elevation, spread that is accounted for in the standard error of the regression. We therefore rephrase this section as follows:

"The remaining elevation differences (hereinafter referred to as elevDiff) are due to time-dependent elevation change that can be related to glacier thickness change as well as errors in the two data sets, temporal heterogeneity (**TanDEM-X is a composite of acquisitions from different years**) and differences in penetration between the reference

DEM (**X-band**) and the swath elevation measurements. **The errors related to the reference DEM will result in an increase in spread of the elevDiff measurements and is accounted for in the regression model discussed below.**"

b. Density conversion and snow accumulation: The author's apply a constant value of 850 kg/m$^3$. This value is often applied also in other studies but needs to be applied with caution. First of all Huss (2013) states that the conversion factor can be significantly different for short periods. This is especially important as height changes of snow which has a much lower density can be large. Hence, the authors need to consider these density variations more carefully especially when interpreting short term changes.

Answer: Assuming a volume to mass conversion factor of 850 ± 60 kg m$^{-3}$ is appropriate for a wide range of conditions and longer term trends, however, this factor can differ significantly for shorter term periods (<3 years) (Huss, 2013). In this study we provide time-dependent elevation changes on a monthly basis, however, caution is required converting these elevation change time series to mass changes.

We added this point in the mass balance sub-section of the method section:

"To obtain volume changes we use the glacierised area of the Randolph Glacier Inventory (RGI 6.0) (RGI Consortium, 2017). We assume the standard bulk density of 850 kg/m3 (Huss, 2013) to convert volume changes to equivalent mass changes. **This assumption is considered appropriate for a wide range of conditions and longer-term trends, however, this factor can differ significantly for shorter term periods (<3 years) (Huss, 2013).**"

We also change the title of section 2.4 to:

"2.4 Time series of **surface** elevation changes"

c. These and other sources of uncertainties need to be better acknowledged. I have the feeling that the provided uncertainty ranges of 0.03 and 0.07 m w.e./a are clearly too low. I suggest showing the accuracy of the data and results at few selected test sites with independent data (e.g. the HMA DEM, Shean et al. 2017, ICESat-2 data, detailed comparison to high resolution DEMs or other studies for smaller regions and also in-situ measurements, e.g. as available from WGMS).

Answer: As described in the response to reviewer #1 we have added a discussion about uncertainties to the paper:

**"While our uncertainty methods follow existing approaches and our error bounds are similar in magnitude to Brun et al. (2017), Kääb et al. (2012) and Shean et al. (2020) but lower than GRACE-based estimates, several additional potential sources of errors could impact the results, and methods to assess them, not currently available, should be developed. Radar altimetry has been shown to be sensitive to surface slopes, and in particular to slope in the direction of the satellite's flight path, in regions like HMA and GoA this impact will also be seen in the performance of the onboard tracker as for large slopes the system is expected to "lose lock". It is a well-known observation**

**that microwave pulses scatter from the surface as well as the subsurface, which can lead to elevation change bias in regions of historically anomalous melt events (Nilsson et al., 2015); or at seasonal time-scale (Gray et al., 2019). Over most regions however, it has been shown that surface elevation change from CryoSat over annual and pluri-annual time scale are consistent with in-situ, airborne, and meteorological observations (Gourmelen et al., 2018; Gray et al., 2015, 2019; McMillan et al., 2014a; Zheng et al., 2018). Using static glacier masks can also lead to errors in regions of rapid dynamic changes. In general these limitations are known and efforts are currently underway in the community to improve uncertainty analysis, and develop new glaciers outlines products."**

And also answered specific comments from reviewer#1 that are relevant here.

The datasets mention by the reviewer are not ideal for validation, the HMA DEM is a static dataset, IceSat-2 only overlaps a few months with the time period of this paper and would deserve a dedicated study as the 2 datasets have a very different spatial sampling. Shean et al. (2020) and Brun et al. (2017) cover the time periods 2000–2016 and 2000–2018 and do not provide estimates on their sub-periods. The difficulty with *in situ* measurements list in the very different spatial scales. In general we do agree with the reviewer that this is a key point that is lacking in regions like the himalaya and dedicated validation study should be carried out.

2. Small glaciers and data coverage: The authors state that CryoSat-2 are also able to survey (very) small glaciers, but do not clearly state a size threshold what they consider as small. There are many very small glaciers especially in HMA which can have significant impact on the overall and especially specific mass changes. Please define "small" and show the portion of the size classes covered in comparison the glacier inventory. Moreover, be more specific about the data coverage and the representativeness and show a plot of the data coverage in relation to the total area.

Answer: We have added the total glacierised area vs glacier size in Figure S5b and S5c (see below). In addition, we have quantified the data coverage and glacierised area for (glaciers smaller than 1km$^2$, glaciers between 1 and 10 km$^2$, glaciers between 10 and 100 km$^2$ and glaciers larger than 100 km$^2$).

[Figure]

**Figure S6:** Relation between glacier size and CryoSat-2 swath elevation data coverage in High Mountain Asia and the Gulf of Alaska region (a), hypsometry of glacier sizes in the Gulf of Alaska (b) and in High Mountain Asia (c). For High Mountain Asia we achieve a data coverage of 25% for glaciers smaller than 1 km² (representing 21% of total glacierised area), 34% coverage for glaciers between 1 and 10 km² (representing 39% of total glacierised area), 37% coverage for glaciers between 10 and 100 km² (representing 28% of total glacierised area) and 31% coverage for glaciers larger than 100 km² (representing 12% of total glacierised area). For the Gulf of Alaska we achieve a coverage of 31% for glaciers smaller than 1 km² (representing 8% of total glacierised area), 42% coverage for glaciers between 1 and 10 km² (representing 15% of total glacierised area), 54% coverage for glaciers between 10 and 100 km² (representing 21% of total glacierised area) and 68% coverage for glaciers larger than 100 km² (representing 56% of total glacierised area).

3. There is no mention of impact of glacier surges and avalanche-fed glaciers which are common in parts of the study regions. The validity of the data for these glacier types should be in an ideal case shown, but at least discussed.

Answer: Dynamic processes such as surges impact our results in two ways. Firstly, we acknowledge that static glacier masks do not capture the temporal evolution of glacier extents induced by dynamic processes, which is one of the limitations of this study, as it currently is for many other studies (described in the section 2.2). To address this issue we recommend using dynamic glacier masks as a step into the future to sample fast changes more accurately. To account for errors due to temporal changes in glacier extents and polygon digitization we use an error of 10% (Shean et al., 2020) on the glacier masks, even though the reported uncertainty of the RGI is ~8% (Pfeffer et al., 2014) (described in the Supporting Information).
Secondly, the hypsometric curves of surging glaciers can differ largely from other surrounding glaciers (Huber et al., 2020), which can affect the results when interpolating surging glaciers with hypsometric averaging on a regional or sub-regional scale. Typically this is addressed by separating surging and non-surging glaciers for interpolation (e.g. Morris et al., 2020), or hypsometric averaging for individual glaciers (e.g. Larsen et al., 2015). In this study we do not achieve the spatial resolution to address this, and we therefore do not distinguish between surging and non-surging glaciers when applying hypsometric averaging. However, it has to be noted that the affected area is relatively small; In this study we only interpolate 4% of glacierised area for the Gulf of Alaska and 12% of

glacierised area for High Mountain Asia. Assuming that we statistically sample dynamic processes with Swath measurements, the impact on our overall elevation change measurements is therefore minimal.

4. The authors exclude the endorheic basis when considering the contribution to sea level rise. Here, the authors need to be more specific: Basins where part of the glacier melt as led to lake level rise (e.g. Neckel et al., 2014) this is suitable, but for the others the water (if not sored in the ground) would end up in the hydrological cycle and ultimately in the oceans.

Answer: In the same manner as Brun et al. (2017) we provide estimates including all glaciers and estimates excluding the major endorheic basins. We have clarified this in the following sections:

In the data methodology section (2.3):
To generate the contribution to sea level rise (SLR) we assume an area of the ocean of 361.8 · 106 km$^2$ and **consider total contributions from all glaciers and then only those glaciers within exorheic basins in High Mountain Asia, based on the HydroSHEDS dataset (Lehner et al., 2006)**.

In the result section (3.2):
"The total HMA mass balance between 2010 and 2019 was –28.0 ± 2.4 Gt yr$^{-1}$ (–0.29 ± 0.03 m w.e. yr$^{-1}$), **or –18.3 ± 1.6 Gt yr$^{-1}$ when including only exorheic basins. This mass loss corresponds to 0.078 ± 0.007 mm yr$^{-1}$ SLE, or 0.051 ± 0.005 mm yr$^{-1}$ when including only exorheic basins.**"

In the conclusion:
"We find that between 2010 and 2019, HMA has lost mass at rates of 28.0 ± 2.4 Gt yr$^{-1}$ (0.29 ± 0.03 m w.e. yr$^{-1}$), and the GoA region has lost mass at rates of 76.3 ± 5.6 Gt yr$^{-1}$ (0.89 ± 0.07 m w.e. yr$^{-1}$), **for a sea-level contribution of 0.078 ± 0.007 mm yr-1 (0.051 ± 0.005 mm yr$^{-1}$ from exorheic basins)** and 0.211 ± 0.016 mm yr$^{-1}$ respectively for HMA and the GoA."

5. Climatic consideration: The authors explain some of the variation by accumulation type and changes in weather and climate. While I agree that this is in principle true the relation to the weather and climate is too simplified. E.g. there are regions in the Tien Shan which receive more accumulation during summer and winter snowfall is also of high importance for parts central Himalaya. Please be here more specific. I suggest to consider more references (e.g. Maussion et al. 2014) and consider Sakai and Fujita (2017) more in detail.

Answer: Thanks for this valuable input. We agree with the reviewer and have rewritten the discussion section on temporal variability in High Mountain Asia:

**"The seasonal and annual time series variability reflects the influence of atmospheric circulations and precipitation seasonality in High Mountain Asia on ice thickness change. Sub-regions dominated by winter accumulation (generally westerly regimes), such as the Hindu Kush, Western Himalaya and the Pamir region (see Pohl et al.,**

**2015; Yao et al., 2012), show the typical seasonal pattern with mass accumulation during winter/early spring and mass losses in the summer/autumn months (Figure 5).**

**Contrarily, sub-regions such as Central Himalaya, Eastern Himalaya and Hengduan Shan show a more heterogeneous seasonal pattern. The elevation change time series of these three sub-regions display that the annual cycle has two peaks, with a first peak in winter and a second and smaller peak in summer (Figure 5, S1). Receiving summer-accumulation through the Indian monsoon these sub-regions generally have a precipitation maximum in July/August, however they are also defined by a high variability of precipitation regimes (Maussion et al., 2014) and a high temperature range (Sakai and Fujita, 2017) resulting in glaciers with varying types over very short distances (Maussion et al., 2014). The impact of this variability becomes evident when compared to the more periodic seasonal patterns of the Hindu Kush, Western Himalayas and Pamir time series. This also stands in contrast with the inner Tibetan Plateau, dominated by a more continental climate, which displays almost no intra-annual cycle.**

**In general, the heterogeneity of the time series reflects the sensitivity of mountain glaciers to meteorological patterns and changes and emphasises that glaciers in High Mountain Asia cannot be considered as one entity with uniform temporal variability and sensitivity to changes."**

6. Sections on main results: Sections 3.2 and 3.3: are important sections as the main results are shown. However, these are really short and lack details. You have much more to show. Highlight here all the important results including the shorter term trends and seasonal variability

Answer: We have extended both result sections (3.2 and 3.3) with paragraphs on seasonal / annual changes and altitudinal distribution of elevation changes and specific sub-regional for both regions. Please refer to answers of the specific comments on L210ff and L213ff for details.

7. Discussion sections: Put more emphasis on possible reasons of mismatches to other studies (also see specific comments).

Answer: We have put emphasis on this in the discussion section in 4.1.3., 4.2.2 and 4.2.3 (for more details see answers in the specific questions). We have extended the discussion on the uncertainties and variability of our own results in sections 2.2, 2.4 and 3.2, which is outlined in detail in the specific comments below and the responses to reviewer #1.

8. HIMAP-regions: In order to be able to better compare the regional results to other studies, I suggest to include also a comparison the subregions presented by the cryosphere chapter of the HIMAP report (Bolch et al. 2019) at least in a figure and table in the supplement. These regions were defined by an international team including local scientists and are often used (e.g. Shean et al. 2020, Rounce et al., 2020). Moreover, these regions capture better the regional variability (e.g. mass balances in East Pamir which are more positive and those

in central and west Pamir which are more negative). However, I do not want to force you as I am one of the lead authors of this HIMAP chapter and leave this decision to you/the editor.

Answer: Thank you for this valuable input. We have included the HiMAP regions in our analysis (see Figure and Table below).

[Figure]

**Figure S9: High Mountain Asia (HMA) specific mass balance trends on a sub-regional level (using the HiMAP sub-regions) in comparison with Shean et al. (2020). This study covers the time period of 2010 to 2019, whilst Shean et al. (2020) cover the time period of 2000 to 2018.**

| | Glacier area [km²] | Specific mass change [m w.e. yr⁻¹] | Mass change [Gt yr⁻¹] |
|---|---|---|---|
| **Eastern Hindu Kush** | 2938 | $-0.27 \pm 0.08$ | $-0.79 \pm 0.22$ |
| **Western Himalaya** | 7986 | $-0.25 \pm 0.06$ | $-2.00 \pm 0.46$ |
| **Eastern Himalaya** | 2844 | $-0.69 \pm 0.11$ | $-1.94 \pm 0.31$ |
| **Central Himalaya** | 8682 | $-0.44 \pm 0.05$ | $-3.84 \pm 0.44$ |
| **Karakoram** | 21472 | $-0.06 \pm 0.02$ | $-1.25 \pm 0.49$ |
| **Western Pamir** | 8418 | $-0.22 \pm 0.05$ | $-1.85 \pm 0.44$ |
| **Pamir Alay** | 1846 | $-0.21 \pm 0.10$ | $-0.39 \pm 0.19$ |
| **Northern/Western Tien Shan** | 2261 | $-0.52 \pm 0.14$ | $-1.17 \pm 0.31$ |
| **Dzhungarsky Alatau** | 521 | $-0.56 \pm 0.09$ | $-0.29 \pm 0.04$ |
| **Western Kunlun Shan** | 8457 | $+0.06 \pm 0.03$ | $+0.51 \pm 0.26$ |
| **Nyainqentanglha** | 7047 | $-0.89 \pm 0.09$ | $-6.29 \pm 0.64$ |
| **Gangdise Mountains** | 1271 | $-0.30 \pm 0.14$ | $-0.38 \pm 0.18$ |

| | | | |
|---|---|---|---|
| Hengduan Shan | 1282 | −0.92 ± 0.24 | −1.18 ± 0.31 |
| Tibetan Interior Mountains | 3815 | −0.10 ± 0.07 | −0.39 ± 0.26 |
| Tanggula Shan | 1841 | −0.42 ± 0.08 | −0.77 ± 0.16 |
| Eastern Tibetan Mountains | 312 | −0.78 ± 0.12 | −0.24 ± 0.04 |
| Qilian Shan | 1598 | −0.30 ± 0.04 | −0.47 ± 0.06 |
| Eastern Kunlun Shan | 2995 | −0.49 ± 0.06 | −1.45 ± 0.17 |
| Altun Shan | 295 | −0.28 ± 0.15 | −0.08 ± 0.04 |
| Eastern Tien Shan | 2333 | −0.45 ± 0.05 | −1.05 ± 0.13 |
| Central Tien Shan | 7270 | −0.31 ± 0.05 | −2.25 ± 0.35 |
| Eastern Pamir | 2118 | −0.22 ± 0.13 | −0.46 ± 0.27 |

**Table S2: High Mountain Asia (HMA) mass balance trends from 2010 to 2019, aggregated on the HiMAP sub- regions (Bolch et al., 2019).**

**Specific comments:**

Title: The title does not fully reflect the content. One of the strengths of the study is that is shows not only one period but annual and the seasonal variability.

Answer: Thank you for this valuable input. We have changed the title to:

"**Spatially and temporally resolved** ice loss in High Mountain Asia and the Gulf of Alaska observed by CryoSat-2 swath altimetry between 2010 and 2019"

Abstract: It is good to keep the abstract short. However, it would benefit if the variations of glacier mass/elevation change found are better highlighted (also in quantitative way).

Answer: We revised this and highlighted the temporal changes in the variations of elevation changes in the abstract:

In the abstract:
"We find that during this period, HMA and GoA have lost an average of −28.0 ± 2.4 Gt $yr^{-1}$ (−0.29 ± 0.03 m w.e. $yr^{-1}$) and −76.3 ± 5.6 Gt $yr^{-1}$ (−0.89 ± 0.07 m w.e. $yr^{-1}$) respectively, corresponding to a contribution to sea level rise of 0.078 ± 0.007 mm $yr^{-1}$ (0.051 ± 0.005 mm yr-1 from exorheic basins) and 0.211 ± 0.016 mm $yr^{-1}$. Glacier thinning is ubiquitous except for the Karakoram-Kunlun region experiencing stable or slightly positive mass balance. **In the GoA region, the intensity of thinning varies spatially and temporally with acceleration of mass loss from −0.03 ± 0.33 m $yr^{-1}$ to −1.1 ± 0.06 m $yr^{-1}$ from 2014 which correlates with the strength of the Pacific Decadal Oscillation. In HMA ice loss is sustained until 2015-6, with a slight decrease in mass loss from 2016, with some evidence of mass gain locally from 2016-17 onwards.**"

L23: Write consistently "Glaciers and ice caps" and include also the percentage area of the glaciers as the area matters more when considering ice melt.

Answer: We have changed this to write consistently "glaciers", which is understood to be land ice outside ice sheets. We have also added the area percentage:

"Glaciers store less than 1% of the mass (Farinotti et al., 2019) **and occupy just over 4% of the area (RGI Consortium, 2017) of global land ice, however their rapid rate of mass loss accounts for almost a third of the global sea level rise, the largest sea level rise (SLR) contribution from land-ice (Bamber et al., 2018; Gardner et al., 2013; Slater et al., 2020; Wouters et al., 2019; Zemp et al., 2019)."**

L40: Include here also Shean et al. (2020)

Answer: We have included Shean et al. (2020)

L43-45: I suggest to include lines 25f here as it is a repetition apart from the seasonality. And please include also a suitable reference for food security and GLOFs.

Answer: To avoid repetition we have moved lines 25f down to this paragraph and re-written the section and added more references.

"**Besides representing an icon for climate change (Bojinski et al., 2014) and impacting global sea level rise, the retreat and thinning of mountain glaciers also affects local communities (Immerzeel et al., 2010). Glacier retreat introduces substantial changes in seasonal and annual water availability, which can have major societal impacts downstream, such as endangering water and food security for populations relying on surface water (Huss and Hock, 2018; Pritchard, 2019; Rasul and Molden, 2019), or introducing geohazards such as extreme flooding (Guido et al., 2016; Quincey et al., 2007; Ragettli et al., 2016). Despite substantial advances with geodetic remote sensing methods, enhancing the spatial resolution and coverage of ice loss estimates, there is currently no demonstrated operational system that can routinely and consistently monitor glaciers worldwide, especially in rugged mountainous terrain and with the necessary temporal resolution.**"

L65ff: I suggest to introduce the subheading "Study regions" here.

Answer:  We have added the subheading study regions.

L66: write ". . . includes the Himalaya, Tibetan mountain ranges, the Pamir and Tien Shan" (Shan=Mountain).

Answer: We have changed this.

L67: write "about 95,500 (or 96,000 or even "almost 100,000") glaciers". Glacier number is a bit arbitrary and depends on the size threshold used and how you split contiguous ice masses.

Answer: We have changed this to "about 95,500 glaciers".

L79: not only since the satellite records but also before. You may then cite here Bolch et al. (2012) which summarised the info for the Himalaya. Please also consider a reference for the Tien Shan which was not covered by the HIMAP report.
.

Answer: We agree with the reviewer that this statement was ambiguous and have rephrased accordingly:

"As a result of atmospheric forcing, the vast majority of glaciers in the HMA region have been **losing mass during the satellite records** (Bolch et al., 2019; Maurer et al., 2019) which has led to widespread glacier slowdown (Dehecq et al., 2019)."

L83: See my comment on the glacier number above

Answer: We have changed this to "about 26,500 glaciers".

L97: See my comment on L79. This applies also here.

Answer: Same as L79 we have rephrased accordingly:

"As a result of atmospheric and oceanic forcings, glaciers in the GoA region have been **losing mass during the satellite records** (Arendt et al., 2002; Berthier et al., 2010; Wouters et al., 2019; Zemp et al., 2019)."

L167: See my comment above. Be more specific

Answer: As described above, we have clarified this in the following sections:

In the data methodology section (2.3):
To generate the contribution to sea level rise (SLR) we assume an area of the ocean of $361.8 \cdot 10^6$ km$^2$ and **consider total contributions from all glaciers and then only those glaciers within exorheic basins in High Mountain Asia, based on the HydroSHEDS dataset (Lehner et al., 2006)**.

In the result section (3.2):
"The total HMA mass balance between 2010 and 2019 was $-28.0 \pm 2.4$ Gt yr$^{-1}$ ($-0.29 \pm 0.03$ m w.e. yr$^{-1}$), **or $-18.3 \pm 1.6$ Gt yr$^{-1}$ when including only exorheic basins. This mass loss corresponds to $0.078 \pm 0.007$ mm yr$^{-1}$ SLE, or $0.051 \pm 0.005$ mm yr$^{-1}$ when including only exorheic basins.**"

In the conclusion:
"We find that between 2010 and 2019, HMA has lost mass at rates of $28.0 \pm 2.4$ Gt yr$^{-1}$ ($0.29 \pm 0.03$ m w.e. yr$^{-1}$), and the GoA region has lost mass at rates of $76.3 \pm 5.6$ Gt yr$^{-1}$ ($0.89 \pm 0.07$ m w.e. yr$^{-1}$), **for a sea-level contribution of $0.078 \pm 0.007$ mm yr-1 ($0.051 \pm 0.005$ mm yr$^{-1}$ from exorheic basins)** and $0.211 \pm 0.016$ mm yr$^{-1}$ respectively for HMA and the GoA."

L188: Omit "very". I would not anymore consider ASTER as "very-high resolution".

Answer: We have changed this.

L206: The Karakoram anomaly was first introduced by Hewitt (2005) and then confirmed by geodetic measurements by Gardelle et al. (2012). Please cite these two references here.

Answer: We have changed accordingly:
"This spatial pattern confirms the suggestion of previous studies (Brun et al., 2017; Gardner et al., 2013; Kääb et al., 2015), that the so-called "Karakoram anomaly" **(Gardelle et al., 2012; Hewitt, 2005)** stretches up to West Kunlun Shan, which has now become the centre of the anomaly."

L207: I am surprised to read about the moderate thinning for Lahaul-Spiti. In line with mass balance measurements and modelling of Chhota Shigri glacier (e.g. Azam et al. 2014), Mukherjee et al. (2018), showed significant mass loss in this region using geodetic data. However maybe mass loss was less after 2010? Please be more specific and quantitative and discuss in the discussion section.

Answer:
Our sub-regional results show significant mass loss (–0.26 ± 0.06 m w.e. yr$^{-1}$) in Spiti-Lahaul, which however are below the average rate for the whole High Mountain Asia region (–0.29 ± 0.03 m w.e. yr$^{-1}$) and therefore described as moderate. Our estimates are in agreement with the –0.31 ± 0.08 m w.e. yr$^{-1}$ of Shean et al. (2020) and the of –0.37 ± 0.09 m w.e. yr$^{-1}$ Brun et al. (2017).

We clarified what "moderate thinning" implies by quantifying this gradient:
"Another striking feature is the gradient from moderate thinning in Spiti-Lahaul and western Himalaya **(–0.25 ± 0.07 m w.e. yr$^{-1}$)** to increasingly negative surface elevation changes along the central **(–0.43 ± 0.05 m w.e. yr$^{-1}$)** and eastern **(–0.56 ± 0.10 m w.e. yr$^{-1}$)** Himalayan mountain range, with the Nyainqêntanglha mountains and Hengduan Shan **(–0.98 ± 0.11 m w.e. yr$^{-1}$)** showing the highest negative trends."

L210ff: Be more specific about the own results here (and double check your results especially considering the uncertainty) and move the critical comparison to other studies to the discussion.

Answer: We have added specific numbers on the sub-regional elevation change rates.

"Our maps of surface elevation change show a heterogeneous pattern in the Himalayan range, with a cluster of slightly positive/near balance trends in the Kunlun and Karakoram ranges (Figure 2), the so-called "Karakoram anomaly" (Gardelle et al., 2012; Hewitt, 2005). **Another striking feature is the gradient from moderate thinning in Spiti-Lahaul and western Himalaya (–0.25 ± 0.07 m w.e. yr$^{-1}$) to increasingly negative surface elevation changes along the central (–0.43 ± 0.05 m w.e. yr$^{-1}$) and eastern (–0.56 ± 0.10 m w.e. yr$^{-1}$) Himalayan mountain range, with the Nyainqêntanglha mountains and Hengduan Shan (–0.98 ± 0.11 m w.e. yr$^{-1}$) showing the highest negative trends.**"

In addition we included results on elevation change at different elevations and seasonal / annual changes:

**"We display the altitudinal distribution of elevation changes in Figure 6 and a comparison with Brun et al. (2017) in Figure S7. While some variability exists along the profiles, in particular over regions and elevation range containing fewer glaciers that can reflect a less robust solution and or spatial variability in glacier response, trends between elevation and ice thickness change are clearly visible. In general, we observe decreasing negative trends with increasing altitudes, which is an expected pattern (Brun et al., 2017; Gardelle et al., 2013). We find the steepest gradient (Figures 6, S7, Table S4) in the Nyainqêntanglha/Hengduan Shan, which is in line with the findings of Brun et al. (2017). We also observe lower or even inverse gradients in Bhutan/East Himalaya, Spiti-Lahaul/West Himalaya, Karakoram/West-Kunlun and Pamir (Figures 6, S7, Table S4), which have been reported previously and been related to debris thickness (Bisset et al., 2020; Brun et al., 2017).**
**We show temporal variability of surface elevation change for the whole HMA region (Figure 4), the RGI second order regions (Figures 5, S1) and the regions by Brun et al. (2017) (Figure S2). The monthly time series show sustained multiannual trends across almost all of the subregions until 2015-6, and decreased loss or even mass gain from 2016/2017 onwards (Figure 5, S2), which is also reflected in the full HMA time series (Figure 4). The Karakoram region in particular displays thinning from 2011 to 2014/5 before abating and thickening again from 2016/7. This shift of thinning rates post-2015 is also clearly seen in Bhutan/East Himalaya, Kunlun (West and East), Tien Shan, Pamir Alay/Hissar Alay and Nyainqêntanglha/Hengduan Shan (Figures 5, S1, S2)."**

L213ff: Similar in this section: Be more specific about the own results and move the critical comparison to other studies to the discussion.

Answer: Similar to the previous item, we have added specific details on the sub-regional elevation change rates.

**"We present sub-regional estimates aggregated on the RGI 6.0 second order regions in Table 2**. The largest mass loss is seen in the Northern Coast Ranges **(–1.08 ± 0.09 m w.e. yr$^{-1}$; –24.8 ± 2.1 Gt yr$^{-1}$)** and Saint Elias Mountains **(–1.03 ± 0.10 m w.e. yr$^{-1}$; –34.1 ± 3.4 Gt yr$^{-1}$)**, especially the Yukutat and Glacier Bay region, which is in line with the spatial patterns of Luthcke et al. (2008) and Luthcke et al. (2013). The lowest thinning rates are observed in the Alaska Range mountains **(–0.41 ± 0.05 m w.e. yr$^{-1}$)**, which is also in agreement with other studies (Berthier et al., 2010; Luthcke et al., 2008)."

In addition we included results on elevation change at different elevations and seasonal / annual changes:

**"We observe a clear correlation between surface elevation changes and altitude (Figure 11, Table S2), with the highest negative trends at low altitudes in the Saint Elias Mountains and Coast Ranges.**
**We display temporal variability of surface elevation change for the whole GoA region (Figure 4), the RGI sub-regions (Figures 9, S3) and for different elevation bands within sub-regions (Figure 10). Figure 9 shows negative trends across all the sub-regions.**

**The four coastal sub-regions – Alaska Pena, Western Chugach Mountains, Saint Elias Mountains and Coast Ranges – display a seasonal oscillation, with an annual surface elevation maximum in spring and annual surface elevation minimum in autumn. In contrast, the seasonal cycle of Alaska Range mountains is shifted, with the thickness maximum in winter, which is also somewhat visible in the time series by Luthcke et al. (2008). A very noticeable feature within the full GoA time series is the steepening (more negative) of inter-annual elevation trends from 2013-4 onwards (Figure 4). We record an acceleration of thinning from –0.03 ± 0.33 m yr$^{-1}$ (before 2014) to –1.1 ± 0.06 m yr$^{-1}$ (2014–2018).  We observe this almost consistently across all the five sub-regions, but this is most pronounced in the Saint Elias Mountains, the Western Chugach mountains and Coast Ranges.**"

L224ff.: This section contains several interesting findings which I suggest presenting in the results section (e.g. the variability in mass changes for the different regions) and keep here, but extend the climatic discussion.

Answer: We agree with the reviewer and have made the following changes:

In the result section (3.2) we added:
"**We show temporal variability of surface elevation change for the whole HMA region (Figure 4), the RGI second order regions (Figures 5, S1) and the regions by Brun et al. (2017) (Figure S2). The monthly time series show sustained multiannual trends across almost all of the subregions until 2015-6, and decreased loss or even mass gain from 2016/2017 onwards (Figure 5, S2), which is also reflected in the full HMA time series (Figure 4). The Karakoram region in particular displays thinning from 2011 to 2014/5 before abating and thickening again from 2016/7. This shift of thinning rates post-2015 is also clearly seen in Bhutan/East Himalaya, Kunlun (West and East), Tien Shan, Pamir Alay/Hissar Alay and Nyainqêntanglha/Hengduan Shan (Figures 5, S1, S2).**"

In the discussion section:
"**The seasonal and annual time series variability reflects the influence of atmospheric circulations and precipitation seasonality in High Mountain Asia on ice thickness change. Sub-regions dominated by winter accumulation (generally westerly regimes), such as the Hindu Kush, Western Himalaya and the Pamir region (see Pohl et al., 2015; Yao et al., 2012), show the typical seasonal pattern with mass accumulation during winter/early spring and mass losses in the summer/autumn months (Figure 5). Contrarily, sub-regions such as Central Himalaya, Eastern Himalaya and Hengduan Shan show a more heterogeneous seasonal pattern. The elevation change time series of these three sub-regions display that the annual cycle has two peaks, with a first peak in winter and a second and smaller peak in summer (Figure 5, S1). Receiving summer-accumulation through the Indian monsoon these sub-regions generally have a precipitation maximum in July/August, however they are also defined by a high variability of precipitation regimes (Maussion et al., 2014) and a high temperature range (Sakai and Fujita, 2017) resulting in glaciers with varying types over very short distances (Maussion et al., 2014). The impact of this variability becomes evident when compared to the more periodic seasonal patterns of the Hindu Kush, Western Himalayas and Pamir time series. This also stands in contrast with the inner Tibetan**

**Plateau, dominated by a more continental climate, which displays almost no intra-annual cycle.**
**In general, the heterogeneity of the time series reflects the sensitivity of mountain glaciers to meteorological patterns and changes and emphasises that glaciers in High Mountain Asia cannot be considered as one entity with uniform temporal variability and sensitivity to changes."**

L249: This is again an important finding and should be move to the results section and the reasoning discussed here

Answer: We have made the following changes.

In the results section:
**"We display the altitudinal distribution of elevation changes in Figure 6 and a comparison with Brun et al. (2017) in Figure S7. While some variability exists along the profiles, in particular over regions and elevation range containing fewer glaciers that can reflect a less robust solution and or spatial variability in glacier response, trends between elevation and ice thickness change are clearly visible. In general, we observe decreasing negative trends with increasing altitudes, which is an expected pattern (Brun et al., 2017; Gardelle et al., 2013). We find the steepest gradient (Figures 6, S7, Table S4) in the Nyainqêntanglha/Hengduan Shan, which is in line with the findings of Brun et al. (2017). We also observe lower or even inverse gradients in Bhutan/East Himalaya, Spiti-Lahaul/West Himalaya, Karakoram/West-Kunlun and Pamir (Figures 6, S7, Table S4), which have been reported previously and been related to debris thickness (Bisset et al., 2020; Brun et al., 2017)."**

In the discussion section (4.1.3):

"We record less mass gain in Kunlun (+0.01 ± 0.03 m w.e. yr$^{-1}$; +0.06 ± 0.03 m w.e. yr$^{-1}$ in the Western part of Kunlun) than previous studies – indicating that the Karakoram anomaly might not persist long-term (Farinotti et al., 2020; Rounce et al., 2020). **This observation is also reflected in the elevation change profile of the Kunlun regions, where Brun et al. (2017) find constant thickening at almost all elevation during the survey time period of 2000 to 2016, whilst we record thinning at lower elevations (see Figure S7). These findings suggest a shift towards negative mass balance at lower elevations in the Kunlun region in comparison to the previous decade.**"

L264: Please avoid the term doubling or almost doubling. There is a clear increase of mass loss, but uncertainty ranges in this studies are (realistically calculated) and large. Consider here also the Study by King et al. (2019) who used similar data for the Himalaya but found less increase.

Answer: We have revised accordingly:

"Besides the differences in data and methodology, a part of these disagreements can be explained by the time periods. Maurer et al. (2019) and **King et al. (2019)** find that the thinning rates in the Himalayas **have increased from the interval 1975–2000 to**

**2000–2016**. This trend seems to have continued in more recent years, with Ciracì et al. (2020) observing an acceleration in mass loss of 10 ± 5 Gt yr⁻¹ per decade for the period of 2002 to 2019, which could explain our more negative mass balance in comparison to Brun et al. (2017) [2000 to 2016] and Shean et al. (2020) [2000 to 2018]."

L296: "Widely discussed and predicted. . ." please provide evidence for that.

Answer: We rephrased accordingly:

"We record less mass gain in Kunlun (+0.01 ± 0.03 m w.e. yr⁻¹; +0.06 ± 0.03 m w.e. yr⁻¹ in the Western part of Kunlun) than previous studies – **indicating that the Karakoram anomaly might not persist long-term (Farinotti et al., 2020; Rounce et al., 2020)**."

L385: Avoid citations in the conclusions (especially when too prominently referring to own work). Move this to the discussion for more details but keep the main statement here.

Answer: We have removed the references:

"**This, along with recent work in the Arctic and Patagonia demonstrates the potential of such a system to monitor trends in ice mass on a global scale and with increased temporal resolution.**"

L305ff: I am not too familiar with all the work from Alaska, but I ask you to be more specific regarding the work you are considering. Some have clear different time periods of analysis. As mentioned above discuss in more detail possible reasons to different results.

Answer: We have added specifics about the time periods of the studies and possible reasons for mismatches in the discussion session:

"Our total mass budget of −76.3 ± 5.6 Gt yr⁻¹ (−0.89 ± 0.07 m w.e. yr⁻¹) agrees with existing estimates, including those using GRACE (−76 ± 4 Gt yr⁻¹ for 2002−2009, −72.5 ± 8 Gt yr⁻¹ for 2002−2019 and −69 ± 11 Gt yr⁻¹ for 2004−2011 by Sasgen et al., 2012, Ciracì et al., 2020 and Luthcke et al., 2013) and ICESat (−65 ± 12 Gt yr⁻¹ for 2003−2010, by Arendt et al., 2013) as well as a study from airborne altimetry (−75 ± 11 Gt yr⁻¹ for 1994−2013 by Larsen et al., 2015) and a consensus estimate combining glaciological and geodetic observations (−73 ± 17 Gt yr⁻¹ / −0.85 ± 0.19 m w.e. yr⁻¹ for 2006−2016 by Zemp et al., 2019) (Figure 8b). Our result is significantly more negative than two GRACE studies, with estimates of −53 ± 14 Gt yr⁻¹ for 2002−2016 (Wouters et al., 2019) and −42 ± 6 Gt yr⁻¹ for 2003−2010 (Jacob et al., 2012). Besides the variations in methodologies and data between these studies, also differences in study area extents, glacier masks and volume to mass conversion factors contribute to the spread of total mass change results. Our estimates correspond to the RGI 1 region (excluding Northern Alaska) to make the results more comparable for future studies. In general, our total mass balance is more negative than most other studies' findings, reflecting the increased thinning rates we show in the sub-regional time series from 2014."

Some remarks on figure 2: The overall pattern of mass changes makes sense and fits to the current knowledge. There are, however, certain 100x100 grids where the mass balance

does not fit. This is especially the case for the central part of Northern Tien Shan in Kazakhstan. There was a positive balance according to your data, but both geodetic data and in-situ measurements of Tuyuksu glacier clearly highlight negative mass balance Kapitsa et al. (2020) and WGMS data. As already mentioned above Lahaul-Spiti has also more negative values in other studies. You may compare to Chhota Shigri glacier which was shown to be representative for the region. In contrast Eastern Pamir and Altun Shan seems more negative than suggested by other studies. This highlights again that a more careful uncertainty analysis and comparison to existing data and studies are needed.

Answer: Thank you very much for this valuable input. Indeed the estimate for this grid cell appears not to be robust as running various regression models generate significant spread of solution. This is also the case for a handful of other grid cells.
We have addressed this issue by adapting a part of the elevation change methodology. In the new approach we use the spread of results from different regression models (using a set of different quality weightings and outlier removal approaches), as a means of identifying grid cells where the results are particularly sensitive to data sampling, data distribution and data weighting. The updated Figure 2 (displayed below) shows negative mass balance in the above-mentioned grid cell, which is now more comparable to other geodetic measurements in this particular grid cell. We retrieve –0.58 ± 0.10 m yr$^{-1}$ in this study between 2010 and 2019, Kapitsa et al. (2020) propose –0.35 ± 0.17 m yr$^{-1}$ for the Tuyuksu glacier between 1998 and 2016, and geodetic measurements by Shean et al. (2020) retrieve an elevation change of –0.40 m yr$^{-1}$ between 2000 and 2018 (no uncertainty given). In addition, the updated results display a more homogeneous pattern on the Tibetan Plateau and in Nyainqêntanglha/Hengduan Shan, which is more comparable to other studies (Brun et al., 2017, Shean et al., 2020). As discussed above, our results in Spiti-Lahaul are within uncertainties with Brun et al. (2017), Shean et al. (2020) and Kääb et al. (2015) (see Figure 7) and the spatial spread is not particularly different here than the results of the named studies.
All other figures, tables and sub-regional estimates have been updated based on the revised elevation change map results. The methodology section has been updated accordingly to include this change in methodology (Section 2.2 and Supplementary Information S1.2).

In the data and methods:
"However, whilst their weights are calculated only according to the power attribute, here we assign each observation a weight based on power and coherence, i.e. measurements with high power and low coherence within the sample will have lower weights assigned (see Supplementary Information S1.1). **We exclude solutions that display extremely large variability across various regression models, considering them as unstable solutions results (see Supplementary Information S1.2)**. When fitting the model, we iteratively exclude measurements that are more than 3σ from the mean distance to the fitted line, until no more outliers are present (e.g. Foresta et al., 2016, 2018). We discard bins that did not fulfil a set of quality criteria based on elevation change uncertainties, temporal completeness, interannual changes and **stability of regression results (see Supplementary Information S1.2). The remaining bins covered more than 96% of the total glacierised area in the GoA region, and 88% in HMA**."

The overall mass change and SLE from the improved dataset is now:

- **HMA:** −28.0 ± 2.4 Gt yr$^{-1}$ (−0.29 ± 0.03 m w.e. yr$^{-1}$), corresponding to a contribution to sea level rise of 0.078 ± 0.007 mm yr-1 (0.051 ± 0.005 mm yr$^{-1}$ from exorheic basins)
- **GoA:** −76.3 ± 5.6 Gt yr–1 (−0.89 ± 0.07 m w.e. yr$^{-1}$), corresponding to a contribution to sea level rise of 0.211 ± 0.016 mm yr$^{-1}$

[Figure]

**Figure 2: Specific glacier mass balance (m w.e. yr$^{-1}$) in High Mountain Asia (HMA) for the period of 2010 to 2019 on a 100 x 100 km grid. The size of the circles is scaled by the total glacierised area within a 100 x 100 km bin.**

---

## Author Response (AR2)

Response to Editor and Reviewers minor comments to:

**Spatially and temporally resolved ice loss in High Mountain Asia and the Gulf of Alaska observed by CryoSat-2 swath altimetry between 2010 and 2019**

Dear Editor,

Thank you for providing us with additional feedback. We have taken the valuable comments on board and modified the manuscript accordingly. Besides stylistic comments and smaller clarifications we have added a separate section explaining the methods used for uncertainty assessment as well as a section discussing uncertainties and limitations.

Yours sincerely,

Livia Jakob

**Reviewer**

**Main comments:**

Overall, the authors addressed the comments thoroughly, so that have only few minor comments left.
Most important is that the manuscript would highly benefit from an explicit section in the main manuscript where you clearly describe how the uncertainty was assessed. The authors may then refer to the supplement for more details if needed. The uncertainty and error sources should then also explicitly discussed in a seperate part of the discussion section. You can then also better argue regarding possible deviations from other studies.

Answer: We have expanded the description on how the uncertainty is assessed in the paper and also added a section about uncertainties in the discussion. See more details in the specific comments (L. 162ff).

**Specific comments:**

L27: The number is true for the past and a specific period. I ask you therefore to use the past tense and be more specific about the period for which this statements is true.The glacier contribution to sea level rise.

Answer: We have changed this accordingly:
"Glaciers store less than 1% of the mass (Farinotti et al., 2019) and occupy just over 4% of the area (RGI Consortium, 2017) of global land ice, **however their rapid rate of mass loss**

**has accounted for almost a third of the global sea level rise during the 21st century (Gardner et al., 2013; WCRP Global Sea Level Budget Group, 2018; Wouters et al., 2019; Zemp et al., 2019)**, the largest sea level rise (SLR) contribution from land-ice (Bamber et al., 2018; Slater et al., 2021)."

L.40: I suggest to update the reference with Immerzeel et al. (2020) as whole HMA is covered.

Answer: We have updated this.

L78: Include an additional reference. Neither Bolch et al. (2019) nor Maurer et al. (2019) cover the Tien Shan.

Answer: We have changed this accordingly:

"As a result of atmospheric forcing, the vast majority of glaciers in the HMA region have been losing mass during the satellite records (Bolch et al., 2019; **Farinotti et al., 2015**; Maurer et al., 2019) which has led to widespread glacier slowdown (Dehecq et al., 2019)."

L. 162ff. This section contains already some results of the uncertainty estimates "our error bounds are similar to …" and reads also partly like a discussion. In this section you should clearly describe what you did and why. While you should discuss the potential sources of errors and uncertainty and the impact in the discussion section. The sentence "In general, these limitations are known and efforts are currently underway in the community to improve uncertainty analysis, and develop new glaciers outlines products" needs some more details and references.

Answer: We agree with the reviewer and have therefore added a separate paragraph about uncertainty calculation in the methods as well as a section on uncertainties in the discussion.

In the methods section:

[revised manuscript text omitted]

L176: Did you consider the +-60 as suggested by Huss et al. in the uncertainty estimate? Yes, but this should be clear from the main manuscript.

Answer: Yes, we considered the ± 60 as suggested by Huss (2013) in the uncertainty estimate. We have clarified this:

"We assume the standard bulk density of 850 **± 60** kg/m$^3$ (Huss, 2013) to convert volume changes to equivalent mass changes."

L. 327: Include a discussion about the deviation of your study and previously published results in Eastern Pamir.

Answer: We have included a discussion and changed the paragraph accordingly:

"Contrasting estimates have also been published for the Pamir and Pamir Alay mountains (Hissar Alay), where high (Kääb et al., 2015), moderate (this study; **Ciracì et al., 2020; Gardner et al., 2013**), slight mass losses (Brun et al., 2017; Shean et al., 2020), and even mass gains (Gardelle et al., 2013) have been reported. **Part of the discrepancy can be attributed to time variability in mass loss (Brun et al., 2017) and driven by fluctuation in winter precipitation (Smith and Bookhagen 2018). CryoSat time series indeed suggest near-balance between 2011 and 2015 and increased mass loss from 2015 onwards, which could account for the higher mass loss estimates in comparison to the DEM differencing studies covering the last two decades (Brun et al., 2017; Gardelle et al., 2013; Shean et al., 2020).**"

**Replies to editor comments**

Note that we have taken all minor stylistic comments on board and do not discuss them in the answers here.

**Specific comments (not including stylistic comments):**

L131 and L145: It is important to clarify over which time period your processing refers

Answer: We have clarified this accordingly:

"Rates of elevation change are then calculated for each 100 x 100 km bin individually based on elevDiff measurements from mid-2010 to mid-2019."

L194: What about the end date? Does this apply to linear rates of change too?

Answer: The end of the time series is in April 2019, with the latest data from June 2019 due to the 90 days window. As opposed to the timeseries, the linear trends start with the earliest data in mid-2010.
We have changed accordingly:

"Note that as opposed to the linear rates the regional and sub-regional time series displayed in this publication start in January 2011 (with the earliest data from November 2010 using the 90 days window), since we retrieve less swath measurements for the first few months of CryoSat-2's life cycle, impacting the quality of the time series pre-2011. The time series in this paper end in April 2019, with the latest data from June 2019 due to the 90 days window."

L265: Clarify the exact start/end dates of these estimates

Answer: We have clarified accordingly:
"We record an acceleration of thinning from –0.06 ± 0.33 m yr–1 (January 2011 to January 2013) to –1.1 ± 0.06 m yr–1 (January 2013 to January 2019)."

L303: The authors need to state that this is in contradiction to their own results using CS2. So their overall value agrees but not the temporal evolution.

Answer: We stated this accordingly:

"Besides the differences in data and methodology, a part of these disagreements can be explained by the time periods. Maurer et al. (2019) and King et al. (2019) find that the thinning rates in the Himalayas have increased from the interval 1975–2000 to 2000–2016. This trend seems to have continued in more recent years, **with Ciracì et al. (2020) observing significant variation in rates of mass loss during the period between 2002 to 2019, with mean rates of loss 35% larger during the CryoSat period than between 2002 and 2010,** which could explain our more negative mass balance in comparison to Brun et al. (2017) [2000 to 2016] and Shean et al. (2020) [2000 to 2018]."

L640: (Figure 3): tell that the authors are splitting up glaciers into different tiles

Answer: We have mentioned this in the result section:

"Surface elevation change maps (Figure 3) display an expected pattern with more negative trends towards lower elevations close to the coast. Note that some of the lower rates observed in the St Elias Mountain are likely the result of the presence of accumulation areas of large glaciers e.g. Hubbard and Bering glaciers in these particular grid cells."

SI L135: Clarify if plot is for a specific site.

Answer: We have clarified this:

"Differences between the TanDEM-X 90m DEM (German Aerospace Center [DLR], 2018) and the swath elevation measurements in Hissar Alay for different along-track and across-track slopes, including median average deviation (MAD) and mean (MEAN) of the elevation differences (referred to as elevDiff)."

---

## Author Response (AR3)

Response to Editors final technical comments to:

**Spatially and temporally resolved ice loss in High Mountain Asia and the Gulf of Alaska observed by CryoSat-2 swath altimetry between 2010 and 2019**

Dear Editor,

Thank you for providing us with final technical comments. We have taken the valuable comments on board and modified the manuscript accordingly.

Yours sincerely,

Livia Jakob

**Replies to editor comments**

**Technical comments:**

L144. check that all citations are made using "(" and not "["

Answer: we have adapted accordingly.

L154. Sorry for being picky on dates but I think this is an important point. Mid-2010 is not super clear to me. Can you provide month instead?

Answer: we have written July 2010 to July 2019

L311. For readability maybe break this sentence into two (after "path")?

Answer: we have changed this.

L316. I am not native so could be wrong here but I feel that "as a function of" reads better. Ignore if I am wrong of course.

Answer: we agree with the editor and have adapted this.

L410. Here again provide month for clarity (January 2011 to December 2015 maybe? or to January 2015?)

Answer: we have specified the exact month.

L755. Sorry for repeating but one of the issues raised by the RAGMAC working group is the need to clarify the start/end dates of any trend or volume change assessment. Could you also apply this to your items (figures and tables) for those readers who may focus on items first before reading the full text? This apply to all figures/table.

Answer: we have added the exact time periods to all figure captions where it was applicable.